# Role of the CTCF binding site in Human T-Cell Leukemia Virus-1 pathogenesis

Ancy Joseph[1], Xiaogang Cheng[1], John Harding[1], Jacob Al-Saleem[2], Patrick Green[2], Malachi Griffith[1], Deborah Veis[1,3], Daniel A. Rauch[1], Lee Ratner [1,4]*

1 Department of Medicine, Washington University School of Medicine, St Louis, Missouri, United States of America, 2 Center for Retrovirus Research and Department of Veterinary Biosciences, The Ohio State University, Columbus, Ohio, United States of America, 3 Department of Pathology & Immunology, Washington University School of Medicine, St Louis, Missouri, United States of America, 4 Department of Molecular Microbiology, Washington University School of Medicine, St Louis, Missouri, United States of America

* lratner@wustl.edu

## Abstract

During HTLV-1 infection, the virus integrates into the host cell genome as a provirus with a single CCCTC binding protein (CTCF) binding site (vCTCF-BS), which acts as an insulator between transcriptionally active and inactive regions. Previous studies have shown that the vCTCF-BS is important for maintenance of chromatin structure, regulation of viral expression, and DNA and histone methylation. Here, we show that the vCTCF-BS also regulates viral infection and pathogenesis *in vivo* in a humanized (Hu) mouse model of adult T-cell leukemia/lymphoma. Three cell lines were used to initiate infection of the Hu-mice, i) HTLV-1-WT which carries an intact HTLV-1 provirus genome, ii) HTLV-1-CTCF, which contains a provirus with a mutated vCTCF-BS which abolishes CTCF binding, and a stop codon immediately upstream of the mutated vCTCF-BS which deletes the last 23 amino acids of the p12 gene, and iii) HTLV-1-p12stop that contains the intact vCTCF-BS, but retains the same stop codon in p12 as in the HTLV-1-CTCF cell line. Hu-mice were infected with mitomycin-treated or irradiated HTLV-1 producing cell lines. There was a delay in pathogenicity when Hu-mice were infected with the HTLV-1-CTCF virus compared to mice infected with either HTLV-1-p12 stop or HTLV-1-WT virus. Proviral load (PVL), spleen weights, and CD4 T cell counts were significantly lower in HTLV-1-CTCF infected mice compared to HTLV-1-p12stop infected mice. Furthermore, we found a direct correlation between the PVL in peripheral blood and death of HTLV-1-CTCF infected mice. In cell lines, we found that the vCTCF-BS regulates Tax expression in a time-dependent manner. The scRNAseq analysis of splenocytes from infected mice suggests that the vCTCF-BS plays an important role in activation and expansion of T lymphocytes *in vivo*. Overall, these findings indicate that the vCTCF-BS regulates Tax expression, proviral load, and HTLV pathogenicity *in vivo*.

**Data availability statement:** The data that support the findings of this study is publicly available from SRA (Sequence Read Archive at the NCBI) with accession number PRJNA1246134.

**Funding:** This work was supported by National Cancer Institute grants to L.R. (R01 CA258359, R21 CA252869) and P.G. (P01 CA100730). The Siteman Cancer Center is supported in part by an NCI Cancer Center Support Grant CA091842. A.J, X.C, J.H., D.R. and L.R received salary support from R01 CA258359 and R21 CA252869. A.J., X.C., J.H., J-A.S., P.G. D.V., D.R. and L.R. received salary support from P01 CA100730. L.R. received salary support from NCI Cancer Center Support Grant CA091842. The funders had no role in study design, data collection and analysis, decision to publish, or preparation of the manuscript.

**Competing interests:** The authors have declared that no competing interests exist.

## Author summary

Human T-cell leukemia virus type 1 (HTLV-1) is a cause of leukemia and lymphoma, and several inflammatory medical disorders. The virus integrates in the host cell DNA, and it includes a single binding site for a cellular protein designated CTCF. This protein is important in regulation of many viruses, as well as properties of normal and malignant cells. In order to define the role of CTCF in HTLV-1 pathogenesis *in vivo*, we analyzed a mutant virus lacking the binding site in humanized mice. We found that this mutation slowed virus spread and attenuated the development of disease. Gene expression studies demonstrated a dynamic role of CTCF in regulating viral gene expression and T lymphocyte activation.

## Introduction

Human T-cell leukemia virus type-1 (HTLV-1) is the cause of adult T-cell leukemia/lymphoma (ATLL) [1]. HTLV-1 is a delta-retrovirus which encodes plus (+) strand classical retrovirus genes, *gag, pol, pr, env*, as well as regulatory genes, *tax* and *rex*, auxilary genes, *p12, p30,* and *p13*, and minus (-) strand gene, *hbz.* The *tax* and *hbz* gene products both have oncogenic activity in tissue culture and mouse models [2,3]. The Tax protein enhances viral and cellular gene transcription, and it has post-transcriptional roles inhibiting apoptosis and DNA repair, and promoting cellular proliferation [3]. Tax is expressed intermittently in a small proportion of ATLL cells at any given time [4,5]. The Hbz protein represses multiple transcriptional pathways, whereas the *hbz* RNA promotes T-cell proliferation [2]. Hbz is expressed continuously by most ATLL cells, and the Hbz protein is critical for viral persistence and disease development [6].

Most ATLL cells have a single copy of the provirus integrated at a wide variety of different chromosomal sites [7]. The 5'portion of the integrated provirus is heavily DNA methylated with histone post-translational modifications consistent with epigenetic silencing [8]. In contrast, the 3'portion of the provirus exhibits little DNA methylation and has characteristic histone modifications of open chromatin. At the border is a binding site for the chromatin barrier element known as 11-zinc finger protein or CCCTC-binding factor (CTCF). There is a single viral CTCF-binding site (vCTCF-BS) in HTLV-1, which is conserved in almost all HTLV-1 strains and other delta-retroviruses [9–11]. In contrast, there are about 55,000 CTCF-binding sites in the cellular genome [12]. CTCF has been shown to be a chromatin insulator resulting in transcriptional suppression or activation, as well as DNA looping activity [13]. The latter is mediated through binding to the cohesin complex [14]. CTCF is important for regulation of latency, replication, and pathogenicity of many DNA viruses, including Kaposi sarcoma herpes virus (KSHV), Epstein-Barr virus (EBV), cytomegalovirus (CMV), herpes simplex virus (HSV), and adenovirus [15].

Studies of cell lines and primary cells infected with HTLV-1 *ex vivo* showed that the vCTCF-BS modulates transcription of the viral genome and cellular genes within

several hundred bases of the provirus [8,16,17]. In order to assess the role of the vCTCF-BS in HTLV-1 replication and pathogenesis, we examined the effect of vCTCF-BS mutation *in vivo*, using a humanized mouse model (Hu-mice). For this purpose, we used non-obese diabetic *scid* IL2 receptor gamma c null kit (NBSGW) mice injected intra-hepatic with human cord blood CD34+ hematopoietic stem cells (HSCs) or injected intra-tibial with human cord blood CD133+ HSCs, and allowed to engraft without irradiation [18]. The rationale for use of two different humanized mouse infection models is that they more closely simulate the effects seen in lymphomatous (humanized mice with intrahepatic injection of newborn mice with CD34+ HSCs) and acute leukemic forms of ATLL (humanized with intra-tibial injection of CD133+ HSCs). Infection of these mice at 13–16 weeks of age perturbs human thymic alpha-beta T-cell development, resulting in expansion in the thymus of mature single-positive CD4+ and CD8+ lymphocytes at the expense of immature and double-positive (DP) thymocytes (Fig 1A) [19]. Human lymphocytes from the thymus, spleen, and lymph nodes are activated in this model, with increased expression of nuclear factor kappa-B (NFκB)-dependent genes. These mice manifest hepatosplenomegaly, lymphadenopathy, and lymphoma.

## Results

### Role of vCTCF-BS in HTLV-1 replication in humanized mice

Hu-mice, 13–16 weeks of age, were assessed for levels of human leukocytes in the peripheral blood by FACS analysis with an antibody to human CD45. Mice from each litter with at least 5% human CD45+ cells, were separated into groups based on levels of human CD45+ cells. Mice were then inoculated intraperitoneally with lethally mitomycin-treated human 729B lymphoid cells infected with an HTLV-1 mutant with a premature stop codon in the p12-coding gene that does not affect known functions of p12, 23 codons from the 3'end of the 297 nucleotide long gene (HTLV-1-p12stop) (n = 10) (Fig 1A and S1 Table) [17]. Another group of mice was also infected with HTLV-1 with the same mutation found in the HTLV-1-p12stop virus, as well as an additional mutation that abrogates the vCTCF-BS (HTLV-1-CTCF) (n = 17) [17]. Use of the p12stop mutant virus was a necessary control since the vCTCF-BS overlaps the p12 coding sequence, and mutation of the vCTCF-BS would alter the p12 protein, if it had not been truncated. Hereafter, the p12stop virus was used as a control for all experiments to assess the effects of vCTCF-BS mutation in HTLV-1. Mice were monitored clinically for up to 12.5 weeks after infection, with analysis of blood samples obtained every 2.5 weeks. Eight litters of mice were used to obtain a sufficient number of mice for statistically valid results, with similar numbers, and levels of CD34+ cells in mice within each litter allocated for infection with HTLV-1-WT, HTLV-1-p12stop, or HTLV-1-CTCF (Figs 1 and S1 and S1 Table). Infected mice that showed symptoms of disease such as 20% decrease in body mass, loss of hair or hunched back, or dehydration were necropsied.

Hu-mice infected with p12stop HTLV-1 showed disease development with a median survival of 4.3 weeks, whereas only 35% of the HTLV-1-CTCF infected mice developed disease within 12.5 weeks of infection (Fig 1B). Experiments were terminated at 12.5 weeks, since this time point is three standard deviations beyond that of the mean survival of HTLV-1-p12stop infected mice. Mice infected with HTLV-1-CTCF had lower mean spleen weights (162 vs 302 mg, p = 0.0074; Fig 1C) and lower absolute lymphocyte counts at the time of necropsy (1289 vs 7600 cell/ul, p = 0.03; Fig 1D) than those infected with HTLV-1-p12stop. In addition, the percent of CD4+ per CD45+ cells in the spleen, liver, and bone marrow were significantly lower in HTLV-1-CTCF infected than HTLV-1-p12stop infected mice (Fig 1E). Mice infected with HTLV-1-WT showed median survival rate of 5.0 weeks (S1Ai Fig) and no significant differences were seen in spleen weight, absolute lymphocyte counts, or in the proportion of CD4+ cells in blood and tissues in comparison with HTLV-1-p12stop infected mice (S1Aii-iv Fig).

Mice were also humanized via intra-tibial injection of CD133 hematopoietic progenitor cells, and infected 13–16 weeks later (Fig 1A and S2 Table). Infection with HTLV-1-CTCF (n = 7) resulted in delayed onset of lymphoproliferative disease compared to mice infected with HTLV-1-WT (n = 6) or HTLV-1-p12stop (n = 3) (S1Bi Fig). At the time of necropsy there

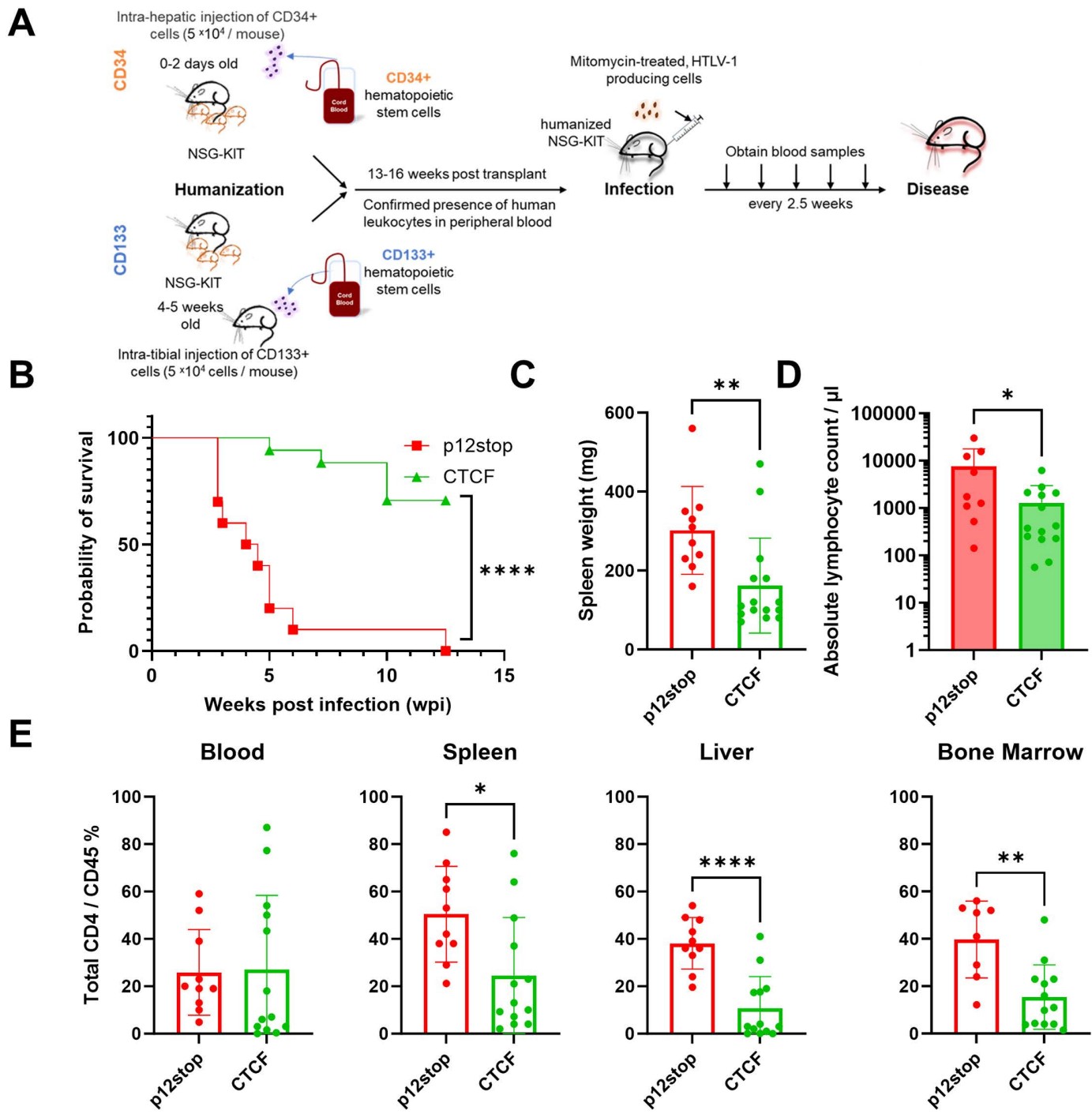

**Fig 1. Infection of CD 34 +humanized mice resulted in decrease in pathogenicity in CTCF Infected mice.** A. Schematic representation of CD34+ and CD133+ hematopoietic stem cell humanization and experimental flow. B. Survival curve of HTLV-1-p12stop, and HTLV-1-CTCF infected CD34+Hu-mice. C. Spleen weights at time of necropsy/death were significantly lower in HTLV-1-CTCF compared to HTLV-1-p12stop infected mice. D. Absolute lymphocyte counts at time of necropsy in peripheral blood. E. Percentage of CD4+T cells among total CD45+cells in blood, spleen, liver and bone marrow cells (* indicates p value lower than 0.05; ** lower than 0.01: *** lower than 0.001; **** lower than 0.0001).(P12-stop-1, CTCF 5, 16, 17 Absolute lymphocyte counts not available due to sample limitation).

were no significant differences in spleen weight (S1Bii Fig), absolute lymphocyte counts (S1Biii Fig), or percentages of CD4+ lymphocytes in the liver in HTLV-1-CTCF compared to HTLV-1-WT infected mice that had been humanized with CD133 hematopoietic cells (S1Biv Fig). However, significant differences in CD4+ lymphocytes were found in the blood, spleen, and bone marrow when compared to combined HTLV-1-WT and HTLV-1-p12stop infected CD133 humanized mice as compared to HTLV-1-CTCF infected mice (S1Biv Fig). Differences in lymphocyte and neutrophil percentages were seen at necropsy in HTLV-1-CTCF compared to HTLV-1-WT infected mice (S2A and S2B Fig). There were insufficient mice infected with HTLV-p12stop for statistical analysis in these experiments. An example of a HTLV-1-WT infected Hu-mouse with ATLL-like flower cells and dramatic leukocytosis is shown in S2C Fig.

Proviral load (PVL) in CD34+ or CD133+ Hu mice blood were assessed by digital droplet PCR every 2.5 wks [20]. The assay measures the number of copies of provirus using primers within the *tax* gene, normalized to number of copies of human ribosomal P subunit p30 gene (Fig 2). At 2.5 wks post-infection, PVL levels in HTLV-1-p12stop infected mice varied between 0.1 and 1.8 copies/cell, with average level of 0.78 copies/cell (Fig 2A and 2B). However, in HTLV-1-CTCF mice, levels were between 0 and 0.02 copies/cell, with average level 0.02 copies/cell that was significantly lower than that in

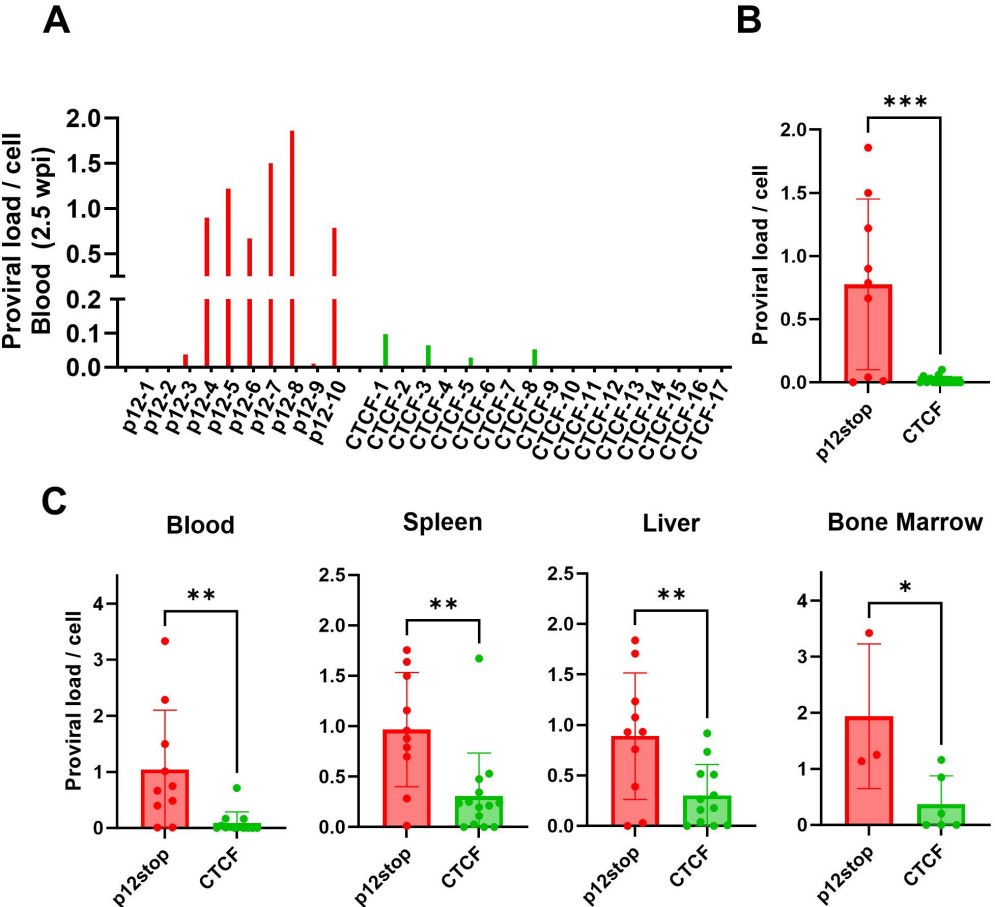

**Fig 2. Repressed proviral loads in CD34+Hu-mice infected with HTLV-1-CTCF.** A. Individual Hu-mice infected with HTLV-1-p12stop had a high PVL at 2.5 wpi, whereas most mice infected with HTLV-1-CTCF had low PVL at all time points up to 10 wpi. B. Comparison of average PVL in HTLV-1-p12stop and HTLV-1-CTCF infected mice at 2.5 wpi. C. PVLs in blood, spleen, liver, and bone marrow cells of infected mice at time of necropsy. (* indicates p value lower than 0.05; ** lower than 0.01; *** lower than 0.001). (PVL for Bone marrow not available for all mice due to sample limitation, please refer S1 Table).

the HTLV-1-WT and HTLV-1-p12stop infected mice (p = 0.0002) (Figs 2A, 2B, S3A and S3B). PVL was also measured at the time of necropsy in blood, spleen, liver, and bone marrow samples (Figs 2C and S3C). In each case, PVL was lower in HTLV-1-CTCF than HTLV-1-p12stop and HTLV-1-WT infected mice. The one exception was that the PVL was lower in HTLV-1-WT than HTLV-1-p12stop infected mouse liver, but the number of animals available for this analysis were small, and this may have been due to a sampling error as a result of heterogeneous levels of virus infected cells within the liver (S3C Fig). Similarly, in CD133+ cell humanized mice, lower proviral loads were seen in HTLV-1-CTCF infected mice compared to HTLV-1-WT infected mice (S4A and S4B Fig). A significant inverse correlation between the proviral load in blood at time of necropsy and survival rate was found in CTCF infected Cd133+ Hu-mice (p value 0.004) (S4C Fig).

## Role of vCTCF-BS in HTLV-1 pathogenesis

The results for HTLV-1-CTCF mice were stratified into two groups, depending upon whether the blood absolute lymphocyte count at time of necropsy was greater than (HTLV-1-CTCF-A; n = 8) or less than 400 cells/µl (HTLV-1-CTCF-B; n = 7; Fig 3A and 3B). Levels of total lymphocytes at necropsy were higher in the blood of HTLV-1-CTCF-A than HTLV-1-CTCF-B infected mice, and similar to those seen in HTLV-1-WT and HTLV-1-p12 infected mice at the time of necropsy (Figs 3E and S5A). In contrast, the number of neutrophils in the blood were higher at necropsy in HTLV-1-CTCF-B than HTLV-1-CTCF-A infected mice (S5A and S5B Fig). Thirty eight percent of the HTLV-1-CTCF-A infected mice died by 7.5 weeks

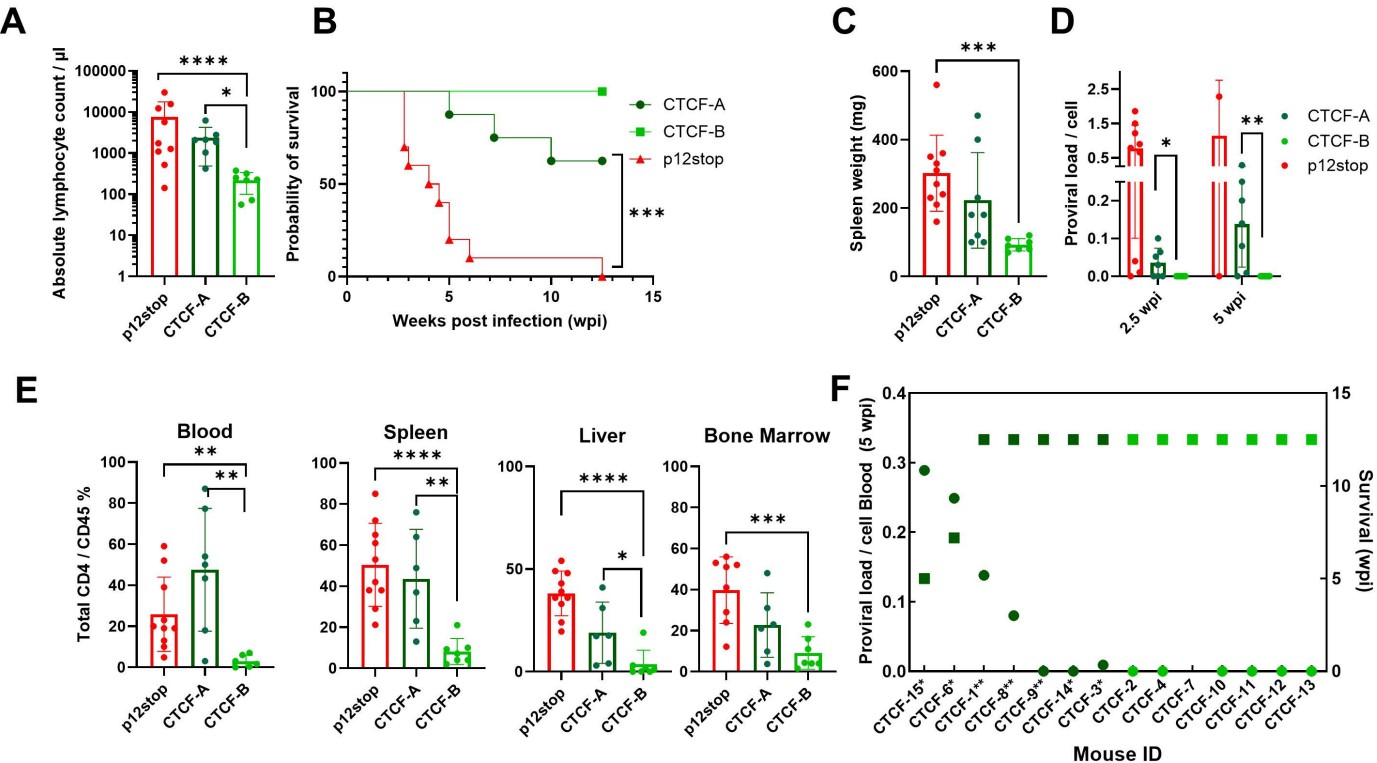

**Fig 3. Comparison of pathogenicity in HTLV-1-CTCF CD34 + Hu-mice based on absolute lymphocyte count.** A. HTLV-1-CTCF infected Hu-mice were separated into two groups based on absolute lymphocyte count at time of necropsy (HTLV-1-CTCF-A infected mice with more than 400 cells/µl; HTLV-1-CTCF-B infected mice with less than or equal to 400 cells/ul) and these groups were compared to HTLV-1P12stop infected mice according to B) survival, C) spleen weight, D) proviral load, and E) percent CD45+ CD4+ in blood, spleen, liver, and bone marrow. F. Correlation of peripheral blood PVL and survival in HTLV-1-CTCF infected Hu-mice. (* indicates p value lower than 0.05; ** lower than 0.01; *** lower than 0.001). CTCF5, CTCF-16 and CTCF-17 were excluded because necropsy could not be performed (S1 and S3 Tables).

post-infection, whereas no disease was seen in HTLV-1-CTCF-B infected mice (Fig 3B). Spleen weight at necropsy (Fig 3C) and PVL was higher at 2.5 and 5 weeks post-infection in HTLV-1-CTCF-A compared to HTLV-1-CTCF-B infected mice (Fig 3D). The ratio CD4+ to CD45+ cells at necropsy in the blood, spleen, liver, and bone marrow were higher in HTLV-1-CTCF-A than HTLV-1-CTCF-B infected mice (Fig 3E). There was an inverse correlation between PVL at 5 wks and survival in HTLV-1-CTCF infected mice (p = 0.0005; Fig 3F). There were no significant differences in levels of human CD45+ cells in animals prior to infection of mice that were subsequently classified as HTLV-1-CTCF-A or HTLV-1-CTCF-B.

Pathological analysis of infected mice that succumbed from infection demonstrated a lymphoproliferative disorder, with diffuse infiltration in the spleen, liver, and lungs (Figs 4A, S6 and S7). The infiltrating cells were found to be predominantly CD4+ lymphocytes, as demonstrated by immunohistochemistry (Figs 4B and S7B). In comparison, control human tissues are shown highlighting CD4+ lymphocytes in normal tonsil, and in biopsies from lymphomatous tissues from two different HTLV-positive patients with ATLL. Furthermore, no pathological abnormalities were seen in HTLV-1-CTCF-B infected mouse tissues (S1 and S3 Tables).

### Effects of vCTCF-BS on transcriptomic profiles

Single cell RNAseq (scRNAseq) analysis was performed on HTLV-1-CTCF-A mice with the highest proviral loads (Fig 5). ScRNAseq analysis performed on a HTLV-1-CTCF-B mouse could not be analyzed due to lack of adequate numbers of human cells. Levels of human cells in splenocytes of HTLV-1-CTCF-B infected mice were 4.5-fold lower than those in HTLV-1-CTCF-A infected mice (Fig 3E).

Viral gene expression was observed in infected splenocytes from 7 animals with lymphoproliferative disease submitted for scRNAseq (S8 Fig). Although 10X scRNAseq reads are not strand specific, splice donor and splice acceptor sites for single-spliced and double-spliced, sense-strand transcripts and for spliced anti-sense transcripts enabled delineation and quantitation of *tax* and *hbz* transcripts (S8A Fig) compared to spliced transcripts of human actin in each sample (S8B Fig). Interestingly, unlike in infected cells in tissue culture [17], *hbz* transcripts were as abundant as plus strand viral transcripts in infected splenocytes *in vivo* (S8A Fig) for both HTLV-1-CTCF (n = 3) and HTLV-1-p12stop infected Hu-mice (n = 4). There were no significant differences in the percent of TCR+ cells that are *hbz*+ in HTLV-1-CTCF infected splenocytes compared to HTLV-1-p12stop infected splenocytes (6.4 ± 3.3 vs 4.6 ± 1.6%, respectively, p = 0.36). The sequence reads also provided confirmation that the nucleotide substitutions used to create the HTLV-1-CTCF and HTLV-1-p12stop viruses were present in 100% of viral transcripts, and that reversion back to the WT sequence did not occur (S8C and S8D Fig).

Analyses of *env* transcripts in infected splenocytes identified the presence of mutations in tryptophan codons resulting in termination codons in 22–56% of reads from HTLV-1-CTCF-A infected mice, but not from HTLV-1-p12stop infected mice (S9 and S10 Figs). Similar mutations in *env* transcripts were also seen at other tryptophan codons at positions 387 and 431 in 1–14% of reads in both HTLV-1-CTCF-A and HTLV-1-p12stop infected mice, and at codons 88, 133, 174, and 438 in 1–6% of reads in HTLV-1-p12stop, but not HTLV-1-CTCF-A infected mice. Analyses of *env* DNA sequences from HTLV-1-CTCF and HTLV-1-p12stop infected splenocytes demonstrated mutations at 1–7% frequency at codon 427, and 4–15% at codon 431, with higher levels seen in at codons 427 (65%) and 431 (61%) in one HTLV-1-CTCF-A mouse each.

Human transcripts, which were clearly separated from murine transcripts, were also analyzed in the scRNAseq results [21]. Although the 729B cells, used as a donor in humanized mice for HTLV-1 infection, contain the EBV genome [22], no EBV transcripts were detectable in the HTLV-1 infected humanized mice at the time of necropsy. Human cells in the spleen of the hematopoietic stem cell transplanted NBSGW mice were exclusively lymphoid cells (S11B Fig). They included clusters of CD4+, CD8+, CD25+, TCR+, and more rarely, NKT, and B lymphocytes. Interestingly, the TCR+ cells of each T cell subset were enriched in T cell activation factors, protein tyrosine phosphatase receptor type C-associated protein (PTPRCAP) and interferon-induced transmembrane protein 1 (IFITM1) (S11 Fig).

At the time of necropsy, reads corresponding to HTLV-1 transcripts were detected in a small subset of splenocytes (Fig 5B). Characteristics of predominant T cell clones, including CD4+, CD8+, Treg, and double CD4+ CD8+ clones, varied

PLOS Pathogens

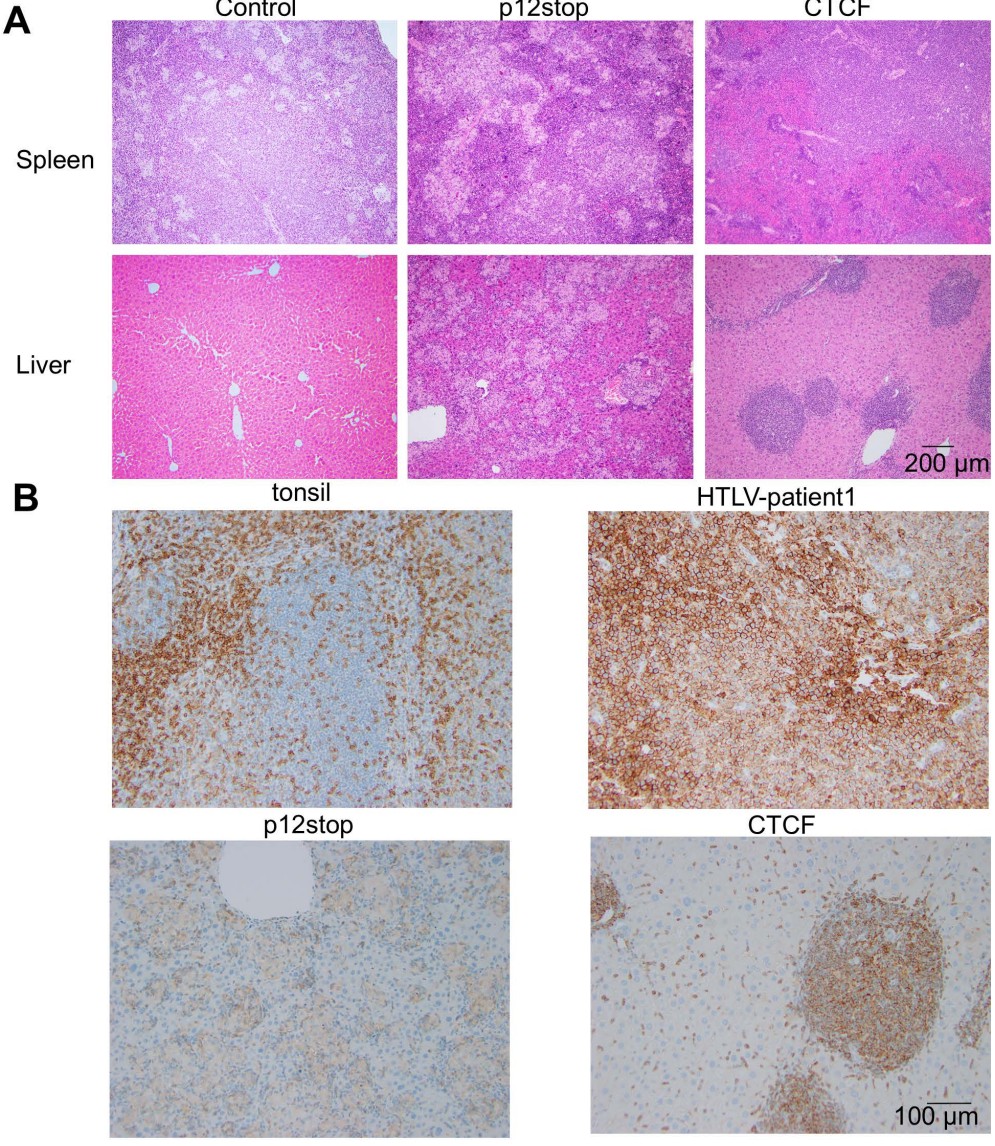

**Fig 4. Histopathological changes in spleen and liver of infected Hu-mice.** A. Hematoxylin and eosin staining (original magnification 10 X) of spleen showing infiltrating lymphocytes in spleen (top panel), and liver showing lymphoid infiltration into the periportal, midzonal and centrilobular region in the liver (lower panel) Infiltration of lymphoid cells are marked with arrow. B. IHC stain for human CD4 inpatient samples as well as humanized mice infected with HTLV-1(original magnification 20 X). Controls for IHC include human tonsil, and sample from a patient who had HTLV-1-associated lymphoma, including a paratracheal lymph node (ATLL patient 1). The lower panel shows CD4 staining in HTLV-1-p12stop-10, and HTLV-1-CTCF-9 infected Hu-mice.

in each sample, and detection of viral transcripts was not restricted to a single T cell clone. T cell clonality was high for all samples, with the Gini coefficient ranging from 0.54 to 0.67, and the Shannon Diversity Index ranging from 3.7 to 5.2 (S12 Fig). There were no significant differences in clonality indices in HTLV-1-p12stop and HTLV-1-CTCF-A infected humanized mouse splenocytes.

Integration site clonality was assessed in infected splenocytes of two HTLV-1-p12stop and two HTLV-1-CTCF-A mice for which sufficient DNA was available. This study examined unique integration sites of the provirus using a novel PCR

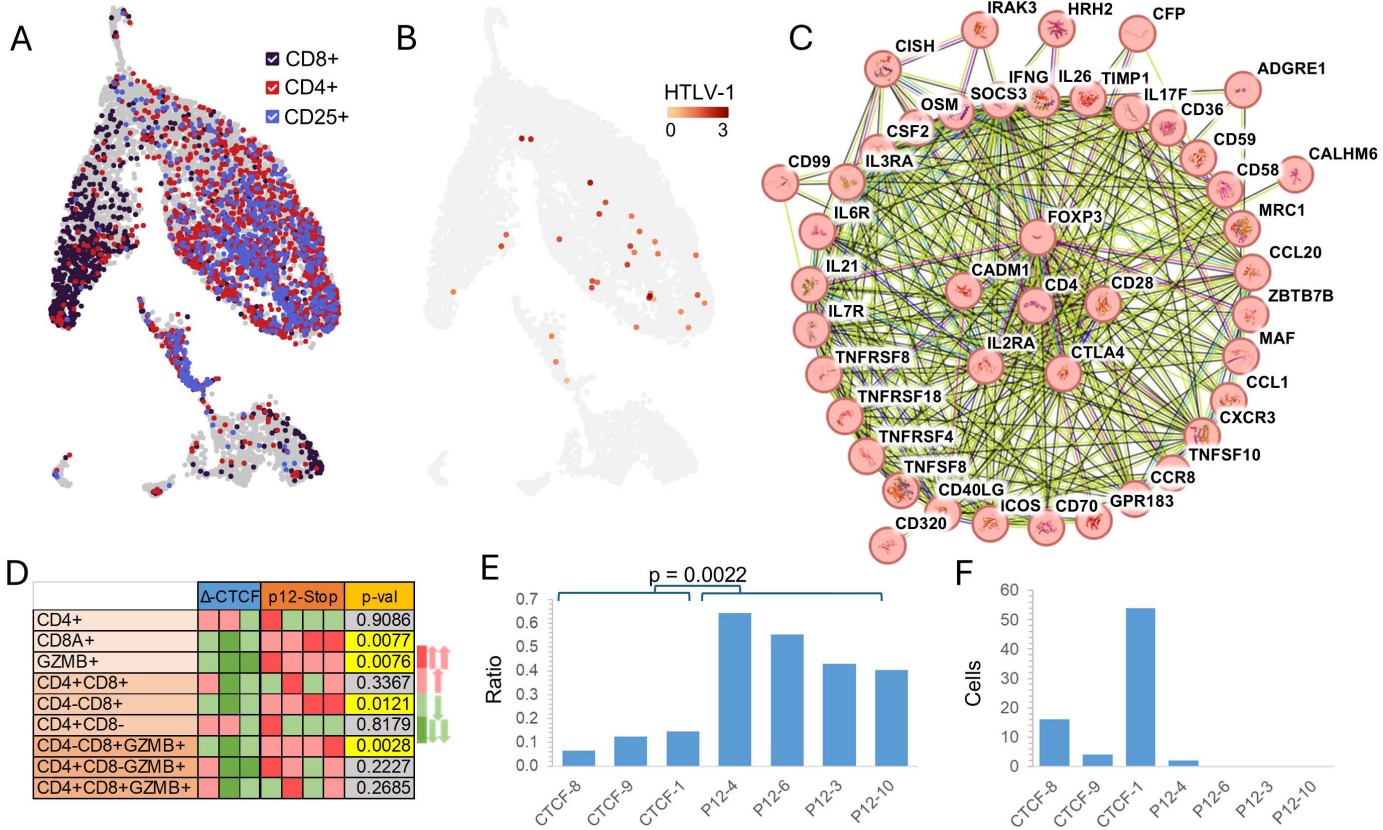

**Fig 5. The vCTCF-BS determines the effect of HTLV-1 on survival and expansion of lymphocytes *in vivo*.** *A*. Single cell RNAseq was performed on splenocytes harvested from Hu-mice infected with HTLV-1-CTCF or HTLV-1-p12stop infected mice. B. The tSNE plots of a representative data set confirm that TCR+ clusters overlap with T cell markers and viral gene expression, and that human CD4+ and CD8+ T cells clusters can be readily identified. C. String network (string-db.org) of genes significantly upregulated in CD4+ T cells confirm that expanded lymphocyte populations in both HTLV-1-CTCF and HTLV-1-p12stop infected Hu-mice express genes associated with ATLL, including IL2RA, FOXP3, BATF3, CD28, and CTLA4. Symbols within circles represent schematic protein structures. D. Heatmap comparing the relative abundance of CD4+ and CD8+ T cells in each sample, normalized against total TCR+ cells in each sample. E. Expression of ALOX5AP shown as the ratio of ALOX5AP+/ALOX5AP- in CD8 vs. CD4 in each sample. F. Number of cells with GLAG peptide in TRB CDR3 of T cell clones.

approach and next generation sequence analysis [23]. In HTLV-1-p12stop infected splenocytes, the Gini coefficients were 0.51-0.91, whereas in the HTLV-1-CTCF infected splenocytes, the Gini coefficients were 0.65-0.7 (S13 Fig). Thus, in both cases, when lymphoproliferative disease is present, a high level of clonality is seen.

String network analysis of genes significantly upregulated in CD4+ T cells demonstrated expanded lymphocyte populations in both HTLV-1-CTCF and HTLV-1-p12stop infected mice (Fig 5C). The cells expressed genes previously associated with ATLL, including CADM1, IL2RA, FOXP3, BATF3, CD28, and CTLA4.

When comparing HTLV-1-CTCF-A infected mice to control HTLV-1-p12stop infected mice, several patterns emerged. First, activated CD8+ T cells were much more abundant in the spleen of HTLV-1-p12stop infected mice (Fig 5D). Granzyme B (GZMB) expressing CD8+ cells were enriched in transcripts for natural killer cell granule protein 7 (NKG7) and granulysin (GNLY), indicating these cells were cytotoxic effector T cells or cytotoxic vesicle releasing cells (S14 Fig) [24]. Second, the expression of 5-lipoxygenase activating protein (ALOX5AP), a regulator of tumor immunity associated with "hot" tumors [25], was a distinguishing characteristic between mice infected HTLV-1-CTCF-A and HTLV-1-p12stop (Figs 5E and S15). Third, double negative CD4-CD8-TCR+ T cells were more abundant in the spleens of HTLV-1-CTCF-A

mice (S16 Fig). Although these cells lacked transcripts for CD4 or CD8, they were enriched in transcripts for calcium-binding helix-loop-helix S100A protein family members (including S100A4, A6, A10, and A11) and interferon-induced transmembrane protein (IFITM) family members (IFITM1 and M2) [26]. Finally, the number of cells in T cell clones carrying a glycine-leucine-alanine-glycine (GLAG) motif in CDR3 of T-cell receptor (TCR) β, previously identified as a Tax-specific epitope [27] was more abundant in HTLV-1-CTCF-A than HTLV-1-p12stop infected mice (Fig 5F and S4 Table). These data suggest that loss of the vCTCF-BS results in significant effects on gene expression and expansion of human T cell populations *in vivo*.

### Effects of vCTCF-BS on temporal viral gene expression

In order to examine the effect of vCTCF-BS on the temporal expression of Tax, we transfected 293T cells with the molecular clones expressing HTLV-1-WT, HTLV-1-p12stop, or HTLV-1-CTCF (Fig 6A). After 48 hrs, these cells produced equivalent quantities of HTLV-1 p19 antigen (5.6±1.2 and 5.4±1.0 ng/ml, in HTLV-1-p12stop and -CTCF expressing 293T cells, respectively). The transfected 293T cells were co-cultivated with Jurkat cells carrying a Tax-responsive red fluorescent protein td tomato (RFP) indicator (JET cells). IncuCyte analysis was performed to assess temporal changes in Tax expression, as measured by RFP fluorescence (Fig 6B and 6C). The number of RFP-positive cells was significantly higher in HTLV-1-p12stop than HTLV-1-CTCF infected cultures from 1-3.5 days of culture, but similar thereafter (Fig 6B). The total RFP intensity was greater in HTLV-1-p12 stop than HTLV-1-CTCF infected cultures from 0-2.5 days of infection, but lower during 2.5-5 days of infection (Fig 6C). Cell viability was assessed using Cytolight rapid dye, and no differences were detected. Similar results were obtained after cocultivation of HTLV-1 infected 729B cells with JET cells (S17 Fig). These results suggest that the vCTCF-BS has dynamic regulation of HTLV-1 gene expression.

### Discussion

Our previous studies of the role of the vCTCF-BS examined in transfected Jurkat cells and PBMCs, the role of mutation of the vCTCF-BS on virus gene expression [17]. We found that mutation of the vCTCF-BS did not disrupt the kinetics and levels of virus gene expression. Furthermore, there was no effect on the establishment or reactivation from latency. Nevertheless, the mutation disrupted the epigenetic barrier function, resulting in enhanced DNA CpG methylation downstream of the vCTCF-BS on both strands of the integrated provirus. We also found enhanced methylation of histones H3 K4, K27, and K36 bound to the provirus.

In our previous study, we also examined the role of CTCF in clonal latently infected Jurkat cell lines carrying the HTLV-1 provirus at different integration sites [17]. For this purpose, we induced viral gene expression with phorbol ester and ionomycin in the presence of a shRNA to repress CTCF expression or a control shRNA. In the majority of these cell lines, knockdown of CTCF resulted in enhanced plus strand gene expression. However, in a minority of cell lines, knockdown of CTCF had no effect on plus strand gene expression. We did not identify cell lines in which knockdown of CTCF decreased virus gene expression. Knockdown of CTCF had no effect on virus gene expression from cell lines with mutation of the vCTCF-BS. Moreover, no effects on minus strand gene expression were seen in any of these cell lines. We found that cell lines manifesting enhanced plus gene expression with CTCF knockdown also exhibited decreased DNA CpG methylation downstream of the CTCF binding site. However, no significant changes were seen in DNA CpG methylation in cell lines not exhibiting alterations of gene expression with CTCF knockdown.

Previously, Martinez et al found that the vCTCF-BS was dispensable for persistent infection in rabbits [28]. However, they noted a decrease in HTLV-1-specific antibody response in rabbits infected with HTLV-1-CTCF. Two significant differences of the rabbit model from the humanized mouse model are that the rabbits are immunocompetent and do not develop high virus load or disease. The contribution of each of these features to differences in pathogenesis in Hu-mice infected with HTLV-1 with or without vCTCF-BS remains to be deciphered.

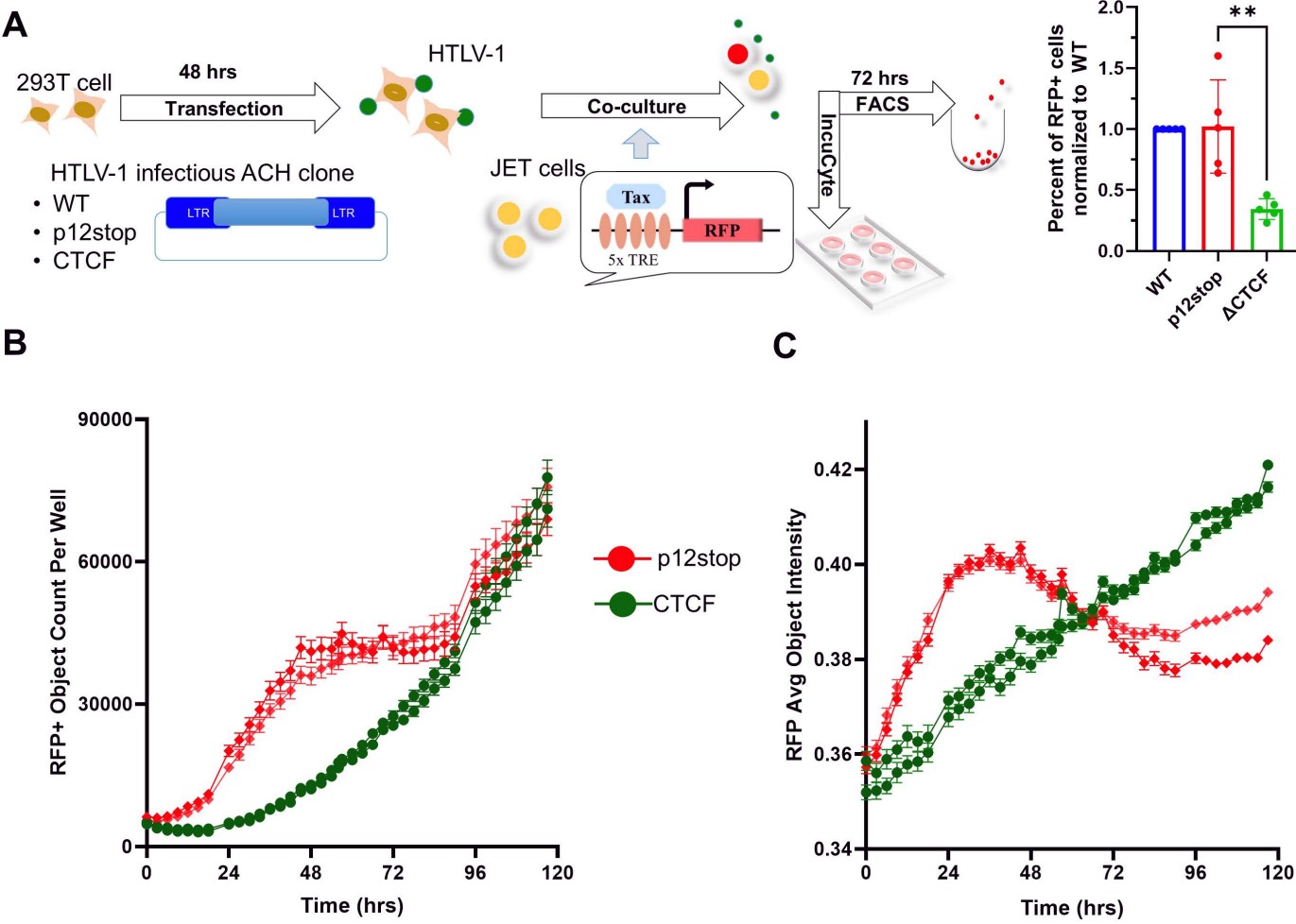

**Fig 6. Temporal expression of Tax in JET cells infected with HTLV-1-CTCF and HTLV-1-p12stop.** A. Schematic flow of the experiment: 293T cells were transfected with pHTLV-1(WT), pHTLV-1(p12stop) or pHTLV-1(CTCF) plasmid respectively, 48 hr after transfection, cells were co-cultured for 72 hrs with JET cells carrying a Tax-dependent td tomato (RFP) indicator for HTLV-1 infection, then RFP positive cells were counted by FACS. The number of RFP positive cells of each infection is presented as a percentage of cells infected with the wt virus (** indicates p value lower than 0.01). To examine the time course of infection, RFP positive cells were monitored for 5 days. (B) Total number of RFP cells (red object) was counted and(C) red mean intensity of infected cells analyzed by using the IncuCyte system. Duplicate samples of HTLV-1-p12stop or HTLV-1-CTCF infected cells are presented at each time point.

In the current work, we examined the effects of the vCTCF-BS mutation *in vivo* in a humanized mouse model [19]. In this model, human CD34+ cells were injected into the liver of newborn mice. After 13–16 weeks, sufficient lymphoid reconstitution occurred to allow HTLV-1 infection, replication, and lymphoproliferative disease. However, we have not detected HTLV-1 antibodies in this model system, suggesting at least partially compromised immune responses to viral infection.

The vCTCF-BS overlaps with the p12 and Hbz coding genes. Mutation of the vCTCF-BS to abrogate binding of CTCF required conservative mutations in these overlapping genes. The p12 mutation truncates the predicted protein product from 99 to 76 amino acids. This truncated protein is similar to that expressed from simian T cell leukemia virus type 1 [29]. Previously, we demonstrated that deletion of the C-terminus of p12 did not affect its ability to functionally enhance nuclear factor of activated T cells (NFAT) [17]. We also showed that the conservative mutation in Hbz had no effect on its ability to repress Tax-mediated viral *trans*-activation or canonical NFκB activity. The mutations in p12 and Hbz used in this study did

not have a significant effect on HTLV-1 replication and pathogenicity in Hu-mice, based on similar results with HTLV-1-WT and HTLV-1-p12stop infected animals.

Mutation of the vCTCF-BS delayed virus spread and delayed or abrogated lymphoproliferative disease in infected Hu-mice (Fig 1). However, the lymphoproliferative disease occurring at late time points in the minority of HTLV-1-CTCF infected Hu-mice were derived from CD4+ lymphocytes as in the case of HTLV-1-WT and HTLV-1-p12stop mice (Fig 4). There was no qualitative change in the characteristics of the lymphoproliferative disease occurring in HTLV-1-CTCF-A infected Hu-mice compared to that present in HTLV-1-WT and HTLV-1-p12stop infected mice.

Single cell RNAseq is a powerful tool for evaluating human lymphocytes within the spleen of infected Hu-mice. The presence of human T cell subsets confirmed that CD34+ hematopoietic stem cells were capable of differentiating into CD4+ and CD8+ T cells *in vivo*. Mature (TCR+) cells were consistently enriched in T cell activation factors PTPRCAP and IFITM1. PTPRCAP is a transmembrane phosphoprotein specifically associated with CD45, a key regulator of T cell activation and differentiation. Along with CD45, CD71, and lymphocyte-specific protein tyrosine kinase (LCK), PTPRCAP (also known as lymphocyte phosphatase-associated phosphoprotein, LPAP) is known to be a major component of the CD4 receptor complex [30]. IFITM1 is a member of a family of interferon-inducible transmembrane proteins that can confer resistance to viral infections, regulate adaptive immunity, and regulate T cell differentiation [26]. Multi-omic evaluation of TCR sequences offered clear evidence of extensive clonal T cell expansion in this model, established that viral gene expression could be detected in expanded clones, and confirmed that the expanded CD4+ T cells were enriched in genes frequently expressed in ATLL cells, including CD25 and cell adhesion molecular 1 (CADM1).

Surprisingly, the most significant difference in the spleens of Hu-mice infected with virus carrying the vCTCF-BS mutation was discovered in the CD8+ T cell population in which the abundance and activity of CD8+ T cells was suppressed relative to control. There were fewer CD8+ T cells, and the CD8+ T cells expressed less Granzyme B and less arachidonate 5-lipoxygenase activating protein (ALOX5AP). Granzyme B is a serine protease and abundant component of cytotoxic granules which when released results in caspase-independent pyroptosis or caspase-dependent apoptosis [31]. Granzyme B is an essential component of immunity and wound healing, and it is also capable of causing injury to healthy tissue or even elevated risk of death [32]. ALOX5AP (also known as 5-lipoxygenase activating protein, FLAP) is required for leukotriene synthesis; it has been implicated in inflammatory responses, stroke, and myocardial infarction; and it is an indicator for predicting high CD8+ tumor infiltration and a "hot" tumor microenvironment [25]. These data establish that scRNAseq can effectively detect human and viral gene expression in mouse spleen in infected Hu-mice, confirm that the expanded lymphocyte populations in this model retain characteristics similar to those described in ATLL, and supports the hypothesis that the vCTCF-BS is involved in viral regulation of immunity and pathogenesis in vivo.

An unexpected discovery from this study was the recurring emergence of viral variants that express envelope transcripts carrying termination codons in the transmembrane subunit. While it is not unexpected that APOBEC3G mediated deamination of TGG can produce TAG stop codons [33], it was striking that the W427* substitution was found in mice in which viral pathogenesis was suppressed. Expression of soluble envelope proteins by a viral reservoir could be a mechanism to block spreading infection and promote latency and host survival [34].

The results of the current study suggest that CTCF binding to the HTLV-1 provirus regulates Tax expression in a time-dependent manner (Figs 6 and 7). Initially, CTCF promotes higher levels of Tax, which results in enhanced plus strand gene transcription, antigen expression, virus production, and enhanced clonal expansion of lymphocytes. However, high levels of Tax and other plus strand genes are associated with enhanced senescence, apoptosis, and immune-mediated responses to virus-infected cells. We conjecture that this results in rapid onset of acute disease in Hu-mice infected with HTLV-1 possessing the vCTCF-BS. In the absence of the vCTCF-BS, there are lower levels of Tax, plus strand gene expression, and virus production, diminished senescence and cytotoxicity, and more gradual lymphocyte expansion resulting in slower development of disease, if disease develops at all. Although mutation of the vCTCF-BS results in an asymptomatic infection or smoldering lymphoproliferative disease in humanized mice rather than an acute

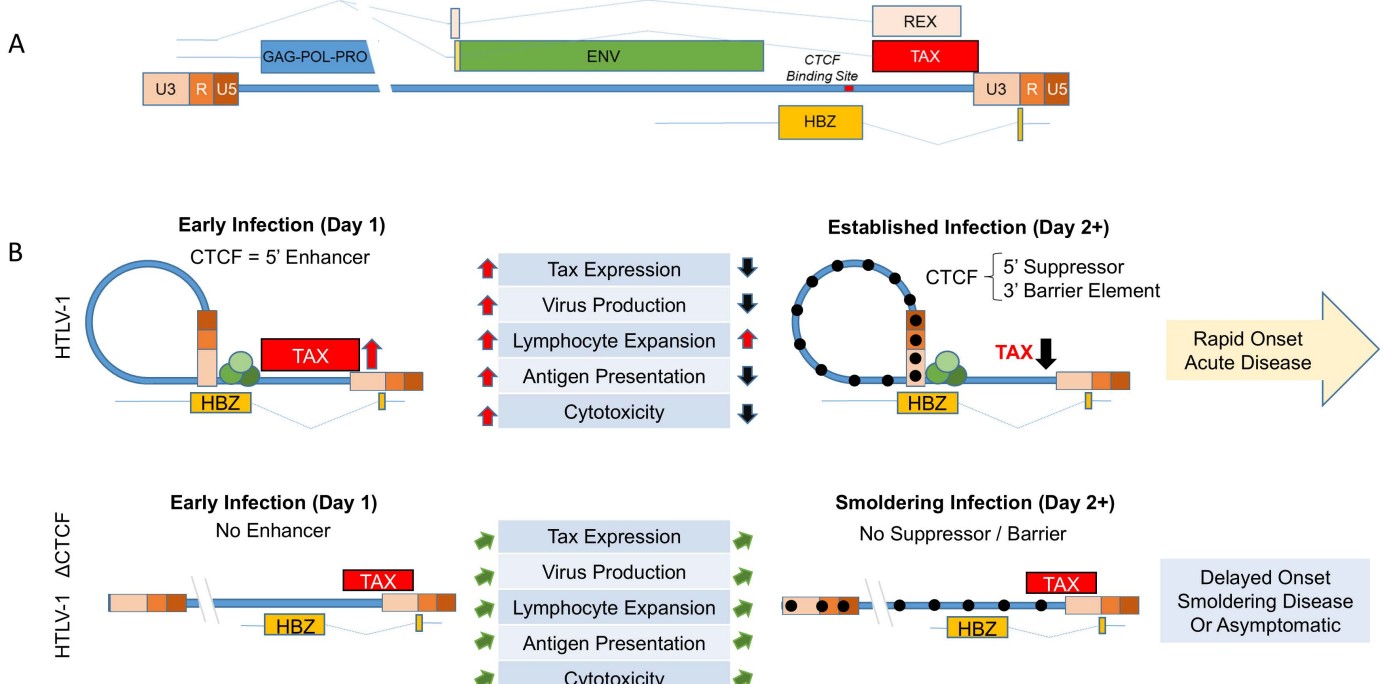

**Fig 7. Working Hypothesis.** A. Schematic depiction of the HTLV-1 genome indicating the location of the vCTCF-BS and major transcripts. B. Working model of the role of the vCTCF-BS in viral gene expression and pathogenesis. Early infection, before methylation of the integrated provirus and suppression of (+) strand transcription from the 5'LTR promoter, the vCTCF-BS acts as an enhancer, Tax expression is elevated, and the effects of TAX protein (virus production, lymphocyte expansion, Tax antigen presentation, cytotoxicity) are elevated. Cytotoxicity resulting from virus production and Tax expression applies selective pressure for (+) sense-strand suppression (DNA hypermethylation) that converts the vCTCF-BS from an enhancer to a barrier element and suppresses Tax expression and activity. This rhythm maximizes virus production early followed by entrance into latency to preserve cellular viability and leads to a burst of initial viremia *in vivo* followed by lymphocyte expansion and death in the infected Hu-mouse with no adaptive immunity (rapid onset acute disease) or a low-level/ undetectable steady state plateau in an infected immunocompetent human (latency). The loss of the vCTCF-BS dysregulates this rhythm, and leads to low level viremia, smoldering infection, and asymptomatic or delayed pathogenesis in the humanized mouse.

leukemic or lymphomatous form of disease, it is premature to suggest that this specific genetic element alone determines the disease phenotype in infected humans.

The mechanism for the effects of CTCF on HTLV-1 transcription could be related to its known silencer effects on initiation or elongation of RNA. This may be a result of monomeric CTCF binding to the provirus or dimeric CTCF-cohesin complexes promoting chromatin looping. High levels of Tax and plus strand transcription at early time points after infection, promote clonal expansion of infected lymphocytes and enhanced viral particle and viral antigen production. Under these conditions, Tax has been shown to induce cellular senescence [35], whereas the viral envelope may induce fusion [36], and multiple viral proteins can induce cell death through direct effects or through immune-mediated cytotoxicity [37]. In contrast, in the absence of CTCF binding to the provirus, there is a more gradual level of Tax expression, resulting in delayed onset or smoldering disease, or asymptomatic infection. It is possible that CTCF activity may contribute to differences in disease subtypes seen in infected patients.

Previous studies by Bangham and colleagues, reported that removal of the vCTCF-BS had no discernible impact on virus transcription or epigenetic modifications in 2 different cell clones of HTLV-1 infected lymphocytes [38]. However, mutation of the vCTCF-BS resulted in altered clone-specific transcription in *cis* at non-contiguous loci up to more than 300 kb from the integration site, suggested to be due to disruption of chromatin loops [16]. Our previous studies contrast

with those of Miura *et al*, in that we examined the effects on viral transcription with loss of vCTCF-BS in a large number of cells *ex vivo* and *in vivo.* In our previous studies with infected rabbits, mutation of the vCTCF-BS did not affect virus replication or spread [38]. However, it is notable that there was a decreased HTLV-1-specific antibody response in this model. Perhaps HTLV-1-specific antibody responses reflect the peak levels of plus strand transcription after infection. It is notable that lymphoproliferative disease does not occur in the rabbit model.

These data suggest a working model in which CTCF regulates Tax, that simulates the *in vivo* survival data, and that HTLV-1-CTCF is less pathogenic than HTLV-1-WT or HTLV-1-p12stop. The scRNAseq gene expression data of 12.5 week old diseased HTLV-1-CTCF-A infected mice look very similar to 5 week old diseased HTLV-1-p12stop mice. The IncuCyte data showed that Tax activity per infected cell is elevated on day 1 and depressed on day 3 in pHTLV-1-p12stop but not HTLV-1-CTCF infected cells, and the number of Tax-positive cells shows rapid expansion in HTLV-1-p12stop but not HTLV-1-CTCF infected mice. Previously published data on latent cell lines show that CTCF is a suppressor of Tax in latent or already suppressed cell clones. This model also explains the observation that although HTLV-1-CTCF and HTLV-1p12stop virus particles are equally infectious, HTLV-1-p12stop infected Hu-mice develop much higher viral loads in the peripheral blood at an accelerated rate compared to HTLV-1-CTCF infected Hu-mice, because the vCTCF-BS present in HTLV-1p12stop infected Hu-mice is an enhancer of Tax and virus production.

A limitation of the current work is that the temporal regulation of Tax expression by CTCF during the first five days of infection was performed in tissue culture models and remains to be confirmed *in vivo*. Such studies require development of highly sensitive assays of virus expression and replication over the course of the first few days of infection *in vivo*. An additional limitation of the study is the requirement for use of irradiated or mitomycin-treated virus-producer cell lines, which although balanced for virus production, infectivity, and immunophenotype, could have more subtle differences in viral or cellular characteristics, that affect virus replication *in vivo.*

In summary, our results demonstrate an important role of CTCF binding to the HTLV-1 provirus in dynamic regulation of virus replication and pathogenicity and support a potentially new discovery that CTCF regulates TAX *in vivo.*

## Materials and methods

### Ethics statement

All the experiments in mice were performed in accordance with ethical and regulatory standards set by NIH for animal experimentation. The animal use protocol (20180321) was approved by Washington University Department of Comparative Medicine. Cord blood samples obtained in this study were obtained from Cleveland Cord Blood Center (CCBC). Verbal informed consent was obtained from all the donors, who were anonymous to the investigators. The study and consent documents were approved by the Washington University Human Research Protection Office.

### Plasmids and sources of cells

The infectious HTLV-1 ACH wild type and mutant clones pHTLV-1(WT), pHTLV-1(p12stop) and pHTLV-1(CTCF) were used in this study [17]. JET cells (JET WT35) and stable 729B HTLV-1 producer cell lines, 729B-HTLV-1(WT), 729B-HTLV-1(p12stop) and 729B-HTLV-1(CTCF) were described in our previous study [17].

Mononuclear cells were isolated by density gradient centrifugation using Ficoll-Paque premium (Sigma Aldrich) and 50 ml SepMate tubes (STEM CELL Technologies) according to the manufacturer's protocol, and CD34+ hematopoietic stem cells (HSCs) were isolated from these mononuclear cells using CD34 microbead kit (Miltenyi Biotec CD34 MicroBead Kit, Human). The purity of isolated CD34+ cells were accessed by flow cytometry using mouse anti human CD34 (BD Bioscience).

### Generation of CD34+ humanized mice (CD34+ HuMice)

NBSGW (NOD.Cg-KitW-41J Tyr+ Prkdcscid Il2rgtm1Wjl/ThomJ), hereafter referred as NBSGW mice were purchased from Jackson laboratories. All mice were kept in animal housing in a pathogen-free environment with ambient temperature,

humidity and controlled light cycles. The NBSGW mice breeding colonies were produced in house. After birth, 0–3 day old pups were injected, using a 27 gauge insulin syringe, intra-hepatically with $5x10^4$ CD 34+ hematopoietic stem cells (HSCs); which were isolated from cord samples collected from full term deliveries (Miltenyi Biotec CD34 MicroBead Kit, Human). Human CD45+ levels were assessed at 13–16 weeks post transplantation (wpt) by flow cytometry analysis.

### Generation of CD133+ humanized mice (CD133+HuMice)

After birth, 4–5 weeks old pups were anesthetized and each mouse was injected with $5x10^4$ CD 133+ hematopoietic stem cells (HSCs), by intra tibial injection, which were isolated from cord blood samples collected from full term deliveries (Miltenyi Biotec CD133 MicroBead Kit, Human). Human CD45+ levels were assessed at 13–16 weeks post transplantation (wpt) by flow cytometry analysis.

### Infection of humanized mice with HTLV-1

Stable 729B HTLV-1 producer cell lines: WT HTLV-1(WT), HTLV-1(CTCF), or HTLV-1(p12stop), which were generated previously [17], were used in this study. Cell lines were maintained in RPMI media (Sigma) supplemented with 10% Fetal Bovine Serum,100 u/ml Penicillin and Streptomycin (Gibco). One million cells/ml were plated in 12 well plates and HTLV-1 p19 antigen in the supernatant was assessed (ZeptoMetrix HTLV p19 Antigen ELISA kit) after 24 hrs. of culture according to the manufacture's protocol. Based on p19 values, cell numbers corresponding to 70 ng p19/mice were used for infection. Before infection HTLV -1 producing cell lines were treated for 90 min with 20 ug/ml of mitomycin C (Sigma Aldrich) to inhibit replication/proliferation of producer cells. Hu-mice were injected with infected cell lines intraperitonially and then were monitored for a period of 12.5 weeks post infection (wpi) for signs of disease. Blood was collected at every 2.5 wpi for proviral load analysis. Mice were anesthetized and necropsied when the body weight dropped by 20% or more of their initial body mass prior to infection. The experimental end point was 12.5 wpi as most of the infected mice showed symptoms by 5 wpi. All the mice were necropsied at the experiment endpoint time (12.5 wpi). Blood, bone marrow, spleen, liver, tumors, and enlarged lymph nodes were collected at the time of necropsy. Complete blood counts (CBC) and Giemsa staining was performed on the peripheral blood smears at the time of necropsy. Infection was confirmed by the presence of detectable PVL in at least one tissue from the humanized mouse.

### Flow cytometry

Peripheral blood was collected by mandibular cheek bleed every 2.5 weeks post infection and at time of necropsy by cardiac puncture after anesthesia with 100 mg/kg ketamine and 20 mg/kg xylazine). Single cell suspensions were made from spleen and liver by, crushing the organs using a wide 1 ml tip and then passing the cell suspension through a sterile 100 µm mesh. PBS supplemented with 2% FBS was used as media. Bone marrow was collected from both femurs by dissection, and then flushing the bones with PBS. All the collected cells were treated with RBC lysis buffer (Sigma-Aldrich) and stained using PE mouse anti-human 45 (BD bioscience) and APC mouse anti-human CD4 antibodies (BD Bioscience). Flow cytometry was performed using BD FACScan (BD Biosciences) and data was analyzed using FlowJo software.

### DNA isolation and proviral load analysis

DNA was extracted from peripheral blood and bone marrow by conventional phenol-chloroform method and Blood and Tissue kit (Qiagen) was used to extract DNA from spleen and liver. A minimum of 50 ng of DNA was used to quantify proviral load. Proviral loads were measured by digital droplet PCR as previously described [17,20].

### Histopathologic analysis

Tissue samples were fixed using neutral buffered formalin (Fisher Scientific) for 24 hours, parafilm embedded and stained with Hematoxylin and Eosin (H&E) solution. To detect the presence of human CD4+ cells, immunohistochemistry was

performed using anti-CD4 (SP35) rabbit monoclonal primary antibody (Ventana Medical systems) according to manufacturer's instruction with slight modification in cell conditioning for 64 min followed by antibody incubation for 40 min at 36 °C. CD4 staining was performed using BenchMark Ultra staining module. Stained sections were observed under a light microscope, and images of whole sections were captured (Nanozoomer) and viewed using the NDP 2.00 viewer (Hamamatsu, Japan).

### JET cell infection, imaging and analysis using IncuCyte system

HEK293T cells were transfected with 2 µg of either pHTLV-1(p12stop) or pHTLV-1(CTCF) using 1 mg/ml polyethylenimine (PEI40K, Polyscience) by a ratio of 3:1 plasmid concentration. After 48 hrs, cells were irradiated (30 Gy) and co-cultured with JET cells [39] and placed in an IncuCyte live cell S3 analysis system (Sartorius). The cells were then continuously imaged for RFP every 3 hrs for 5 days. The IncuCyte software was used to calculate the read mean intensity and total red object count.

### Single cell RNAseq and analysis

Samples of viably cryopreserved mouse splenocytes stored in liquid nitrogen were retrieved immediately before sample processing and submission. Cells were thawed partially in a 37ºC water bath and then placed on ice immediately. Single cell suspensions were revived in ice cold medium by gently adding the cell suspension to 10 ml of RPMI medium supplemented with 10% FBS. Cells were centrifuged and gently washed with PBS with 2% FBS and passed through a 70 µM cell strainer to avoid clumps while processing samples. Cells were stained with 0.4% trypan blue to quantify viability and submitted to the McDonnell Genome Institute for processing for scRNAseq using the 10X Genomics 5' GEX plus TCR V(D)J enrichment. Libraries were prepared using the 10x Genomics 5′ or 3' immune profiling kit-snRNA-seq protocol (GTAC@MGI) and sequenced on an Illumina S4 flow cell. Alignment and gene expression quantification was performed using the CellRanger multi pipeline (v7.1.0). Reads were mapped against the human genome (GRCh38/hg38), the HTLV-1 genome (ViralProj15434; GCA_000863585.1), and the EBV genome (strain B95-8; GCA_002402265.1) and visualized using t-distributed stochastic neighbor embedding (t-SNE) or uniform manifold approximation and projection (UMAP) plots clustered to enable cell type identification. Custom analysis, differential expression, and creation of feature plots was performed using Loupe Browser 8. BAM files generated from CellRanger pipeline were visualized on IGV_2.16.0.

### PCR HTLV-1 *env* gene and DNA sequencing

DNA was isolated from spleen cells of infected humanized mice by phenol-chloroform extraction. HTLV-1 *env* gene was amplified using the forward primer 5'-ATGGGTAAGTTTCTCGCCAC and reverse primer 5'- TTACAGGGATGACTCAGGGT. Amplified DNA underwent gel purification and Sanger sequencing.

### Deep DNA sequencing and bioinformatic analysis

DNA libraries were prepared by following the protocol of Rosewick et al [23]. Briefly DNA was isolated from mouse spleen and sheared by sonication. Fragments containing the junction of HTLV-1 LTR and human genome was initially amplified by a linear extension PCR with incorporation of biotinylated dUTP. Then biotinylated DNA was selected with streptavidin beads and ligated to a partially double-stranded DNA linker. DNA libraries were finally generated by a nest PCR using primers targeting HTLV-1 LTR U5 and the linker sequence. The libraries were sequenced with paired ends and a read length 150 bp.

### Statistics

P values were determined by unpaired t tests. using GraphPad Prism version 10.0.0 for Windows, GraphPad Software (Boston, Massachusetts). The Spearman correlation method was used to determine statistically significant correlation

between proviral load and survival curve. Results were considered to be significant if the p value was ≤ 0.05 (* indicates p value ≤ 0.05, ** for ≤0.01, *** for ≤ 0.001, **** for ≤ 0.0001).

## Supporting information

**S1 Fig. Infection of humanized mice resulted in decrease in pathogenicity in HTLV-1-CTCF infected mice.** Mice humanized with CD34+ cells (A) or CD133+ cells (B) were infected with HTLV-1-WT, HTLV-1-p12stop, and HTLV-1-CTCF viruses for 12.5 weeks or when signs of viral pathogenesis were evident (20% weight loss and lethargy) and evaluated based on; i) survival, ii) spleen weight, iii) absolute lymphocyte count, iv) and human CD4+ lymphocyte count in blood, spleen, liver, and bone marrow. Mice infected with HTLV-1-CTCF had longer median survival, and fewer human CD4+ lymphocytes in spleen and bone marrow than mice infected with HTLV-1-WT or HTLV-1-p12stop in both models. (* indicates p value < 0.05; ** < 0.01; *** < 0.001, **** < 0.0001.)(In S1Aiii-S1Av Fig Wt-1 5, 6, p12-2,3,CTCF5,6,16,17 some data were not included due to less availability of sample, Raw data in S1 and S2 Tables).
(PNG)

**S2 Fig. Infection of CD133+ humanized mice resulted in decrease in pathogenicity in HTLV-1-CTCF infected mice.** Comparison of A) lymphocyte and B) neutrophil percentages in the peripheral blood in HTLV-1-WT, -p12stop and -CTCF infected mice at the time of necropsy. C) Blood smear of HTLV-1-WT infected mouse #5 at 8 weeks of infection with wbc count of 142,000/μl in which flower cells are evident. * indicates p value < 0.05.
(PNG)

**S3 Fig. Repressed proviral loads in CD 34+ Hu-mice infected with HTLV-1-CTCF.** Proviral load calculated as the ratio of proviral DNA to human DNA (quantitated by ddPCR) in peripheral blood obtained at 2.5 weeks post infection A) for each mouse in the study and B) as an average of each group. C) Average proviral load in blood, spleen, liver, and bone marrow cells of each group of infected mice at time of necropsy. * indicates p value < 0.05; ** < 0.01; *** < 0.001(For WT,p12 stop and CTCF mice some PVL not available for bone marrow due to sample limit, Refer to S1 Table).
(PNG)

**S4 Fig. Proviral load in blood was decreased in HTLV-1 infected CD133+ Hu-mice when vCTCF-BS was deleted.** Proviral load calculated as the ratio of proviral DNA to human DNA (quantitated by ddPCR) in peripheral blood of infected CD133 humanized mice obtained A) at 2.5 weeks and 5 weeks post infection and B) time of necropsy as an average of each group. C) A double-Y plot showing A significant correlation between the proviral load on left axis and survival in weeks post infection on the right axis for each CD133+ mouse infected with HTLV-1-CTCF. ** indicates p value lower than 0.01)(For PVL load at week time point in S4A and S4B Figs, some samples data not available due to sample limitation, refer S2 Table)
(PNG)

**S5 Fig. Differential Cell Counts and Proviral loads in CD 34+ Hu-Mice.** At the time of necropsy, complete blood count with differential was performed and proviral loads were determined on blood, spleen, liver, and bone marrow. A) Lymphocyte and neutrophil percentage in two groups of CTCF infected CD34 humanized mice and B). lymphocyte and neutrophil percentages in HTLV-1-WT, HTLV-1-p12stop and HTLV-1-CTCF infected mice. C) Comparison of Proviral loads in blood, spleen, liver in two groups of HTLV-1-CTCFmice with p12 stop infected mice at the time of necropsy. * indicates p value < 0.05; ** < 0.01; **** < 0.0001.(CTCF-5, 16, 17 mice not included in the grouping as we couldn't obtain differential blood count; mouse dead before sample collection).
(PNG)

**S6 Fig. Pathology of spleen, liver in infected CD34 hu-mice at time of necropsy- Enlarged view.** Hematoxylin and Eosin staining (original magnification 20 X) of spleen showing infiltrating lymphocytes in spleen, in which mitosis can be

observed in several cells (upper panel), and liver showing lymphoid infiltration into the periportal/ midzonal/ centrilobular region in the liver (lower panel). Spleen and liver from WT-5, p12-10 and CTCF-9 are shown as representative images to show lymphoid infiltration.
(PNG)

**S7 Fig. Histopathological changes in spleen and liver of infected CD34+ Hu-mice.** A. Hematoxylin and Eosin staining (original magnification 10 X) of spleen showing infiltrating lymphocytes in spleen (upper panel), and liver showing lymphoid infiltration into the periportal, midzonal and centrilobular region in the liver (lower panel). Infiltration of lymphoid cells are marked with arrow. B. IHC stain for human CD4 in patient samples as well as humanized mice infected with HTLV-1(original magnification 20 X). Controls for IHC include human tonsil, and samples from two patients who had HTLV-1-associated lymphoma, including a paratracheal lymph node (HTLV-patient 1) and sphenoid mass (HTLV-patient 2). The lower panel shows CD4 staining in HTLV-1-WT-5, HTLV-1-p12stop-10, and HTLV-1-CTCF-9 infected Hu-mice.
(PNG)

**S8 Fig. Viral gene expression in human cells in spleen of CD34 humanized mice infected with HTLV-1-p12stop and HTLV-1-CTCF.** Bam files visualized in IGV of reads from CellRanger 10x scRNAseq mapped to A) the HTLV-1 genome or, B) the human genome (ACTB gene). Samples of splenocytes were submitted for scRNAseq in two batches (indicated by name colors). Reads spanning splice junctions are indicated by arcs. Numbers for each arc indicate depth of splice-junction spanning reads. C) Sequence of HTLV-1-WT, HTLV-1-p12stop, and HTLV-1-CTCF virus at positions 6708–6737 of the viral genome and D) actual reads of viral transcripts expressed from infected splenoyctes demonstrating that 100% of expressed transcripts carry the correct nucleotide sequence and no reversion has occurred in vivo.
(PNG)

**S9 Fig. Replacement of Tryptophan with Stop Codons in Envelope Transcripts in Infected Splenocytes from CD34 hu-mice.** A) HTLV-1 proviral genome with magnification of the envelope gene (ENV; gp62) composed of the surface (SU; gp46) and transmembrane (TM; gp21) subunits and including the N-terminal signal peptide (S), the fusion peptide (F), the membrane spanning domain (T), and the cytoplasmic tail (C). The HTLV-1 envelope gene contains 13 tryptophan (W) residues, a cluster of three of which (W427, W431, W438) are near the membrane spanning region (G446-L464) of TM. B) Read coverage visualized using IGV of scRNAseq transcripts obtained from splenocytes of infected humanized mice show TAG (stop) codons in transcripts from CTCF-infected mice at position W427. Visualization of allele frequency threshold set at 0.2 These reads would be predicted to result in a truncated envelope protein lacking the membrane spanning region and the cytoplasmic tail but retaining the surface exposed receptor binding domains in SU and TM.
(PNG)

**S10 Fig. Replacement of Tryptophan with Stop Codons in HTLV-1 DNA and RNA Obtained from Infected Splenocytes from CD34 hu-mice.** Percent of HTLV-1 envelope transcripts (RNA reads detected by scRNAseq) or HTLV-1 envelope proviral sequences (DNA reads detected by PCR and sanger sequencing) in infected splenocytes in which tryptophan residues are replaced with termination codons.
(PNG)

**S11 Fig. TCR+ Cells Express PTPRCAP and IFITM1.** A. Volcano plots comparing gene expression in TCR+ cells to TCR- cells for two representative samples. Data points representing PTPRCAP and IFITM1 are shown in color. B. TSNE plots showing enrichment for PTPRCAP and IFITM1 among the CD3+TCR+ population of cells. C. Table of average reads per cell of PTPRCAP and IFITM1 in CD3+TCR+ cells vs. CD3+TCR- cells in each sample with corresponding fold change and p value.
(PNG)

**S12 Fig. Clonality Analysis.** A) Calculation of the Gini index of the cumulative proportion of cells per clone in the cumulative proportion of clones for B) Each T cell subtype in each sample, or C. Each sample as a whole compared to the Shannon Diversity Index for each sample.
(PNG)

**S13 Fig. Clonality analysis of infected mouse cells.** DNA was isolated from the spleen of humanized mice infected with HTLV-1. Unique integration sites and GINI index were calculated after deep sequencing. Each integration site was indicated with different color on the pie graphs and the size of each piece represents the clone size (only showing the number of major clones). P12-4, P12-6, P12-3, P12-10 and CTCF-1, CTCF-8, CTCF-9 are two groups of mice infected with either P12 (wt control) or CTCF mutant virus respectively.
(PNG)

**S14 Fig. Gene Expression in CD8 T cells.** A) Volcano plots and B) String analysis of genes enriched in human CD8+T cells among splenocytes in infected CD34 hu-mice.
(PNG)

**S15 Fig. ALOX5AP Expression.** A) The number of ALOX5AP+ and ALOX5AP- cells among TCR+CD4+ and TCR+CD8+populations in each sample; the ratio of ALOX5AP+ to ALOX5AP- cells for each sample; normalized to CD4 for each sample; also depicted in graphical form. B) tSNE plots for the same ALOX5AP populations in each scRNAseq sample.
(PNG)

**S16 Fig. Double Negative (DN) T cells Abundance and Gene Expression.** A) Table and corresponding graph indicating the number of TCR+CD4-CD8- cells in each sample. B) Volcano plots of gene expression in DN T cells in two representative samples.
(PNG)

**S17 Fig. Variation of temporal expression of Tax in JET cells infected with HTLV-1-CTCF and -p12stop.** 729B cells producing equal amounts of HTLV-1-p12stop, or HTLV-1-CTCF virus were co-cultured with JET cells, carrying a Tax-dependent td tomato (RFP) indicator. In order to examine the time course of infection, co-cultured cells were placed in the IncuCyte and were observed for 5 days. A. The total red object count is shown for cells infected with HTLV-1-p12stop or -CTCF. B. The total red intensity per object in infected cells is shown. C. Graph showing infectivity of stable clones of 729B cells with HTLV-1-p12stop, or HTLV-1-CTCF when co-cultured with JULR cells which carries a luciferase reporter gene. Infectivity was measured after 48 hrs of co-culture by luciferase assay. JET cells co-cultured with JULR was used as negative control.(Error bar represents standard error in S17A and S17B Fig).
(PNG)

**S1 Table. Raw data for all CD34+humanized mice.**
(PDF)

**S2 Table. Raw data for all CD133+humanized mice.**
(PDF)

**S3 Table. Raw data for CTCF infected mice grouped into HTLV-1-CTCF-A infection, or having absolute lymphocyte count greater than 400 cells/µl and HTLV-1-CTCF-B less than or equal to 400 cells/µl.**
(PDF)

**S4 Table. TCR V(D)J sequencing and subtyping.** Columns include the number of individual cells within each clone for the ten most abundant clones in each sample; the V(D)J and CDR3 read for TRB and TRA for each clone; the T cell subtype of each clone (Treg, CD4, CD8, Double Positive); and the number of HTLV reads detected per clone. (PDF)

## Acknowledgments

We thank the Alvin J. Siteman Cancer Center at Washington University School of Medicine and Barnes-Jewish Hospital in St. Louis, MO. and the Institute of Clinical and Translational Sciences (ICTS) at Washington University in St. Louis, for the use of the Genome Technology Access Center, which provided scRNAseq services and support. ICTS is funded by the National Institutes of Health's NCATS Clinical and Translational Science Award (CTSA) program grant #UL1 TR002345.

## Author contributions

**Conceptualization:** Lee Ratner.

**Data curation:** Ancy Joseph, Xiaogang Cheng, Malachi Griffith, Deborah Veis, Daniel A. Rauch.

**Formal analysis:** Ancy Joseph, Xiaogang Cheng, Malachi Griffith, Daniel A. Rauch, Lee Ratner.

**Funding acquisition:** Patrick Green, Lee Ratner.

**Investigation:** Ancy Joseph, Xiaogang Cheng, John Harding, Malachi Griffith, Deborah Veis, Daniel A. Rauch, Lee Ratner.

**Methodology:** Ancy Joseph, Xiaogang Cheng, John Harding, Jacob Al-Saleem, Patrick Green, Malachi Griffith, Deborah Veis, Daniel A. Rauch, Lee Ratner.

**Project administration:** Patrick Green, Daniel A. Rauch, Lee Ratner.

**Supervision:** Xiaogang Cheng, Patrick Green, Daniel A. Rauch, Lee Ratner.

**Validation:** Ancy Joseph, Xiaogang Cheng, Malachi Griffith, Deborah Veis, Daniel A. Rauch, Lee Ratner.

**Writing – original draft:** Ancy Joseph, Xiaogang Cheng, Daniel A. Rauch, Lee Ratner.

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
