## [Decision Letter · Decision Letter 0]

Dear Lee,

Thank you very much for submitting your manuscript "Role of the CTCF Binding Site in Human T-Cell Leukemia Virus-1 Pathogenesis" for consideration at PLOS Pathogens. As with all papers reviewed by the journal, your manuscript was reviewed by members of the editorial board and by several independent reviewers. In light of the reviews (below this email), we would like to invite the resubmission of a significantly-revised version that takes into account the reviewers' comments.

Thank you for your patience in awaiting feedback on your manuscript. We have now received comments from three reviewers. While all three found your results of interest, all also made a considerable number of comments and suggestions regarding the phrasing of the paper and, more substantially, the interpretation of the experimental results and the extent of the claims made.

We cannot make any decision about publication until we have seen the revised manuscript and your response to the reviewers' comments. Your revised manuscript is also likely to be sent to reviewers for further evaluation.

With best wishes,

Charles

Charles R M Bangham, ScD FRS

Academic Editor

PLOS Pathogens

Richard Koup

Section Editor

PLOS Pathogens

Michael Malim

Editor-in-Chief

PLOS Pathogens

orcid.org/0000-0002-7699-2064

Thank you for your patience in awaiting feedback on your paper. We have now received comments from three reviewers. While all three found your results of interest, all also made a considerable number of comments and suggestions regarding the phrasing of the paper and, more substantially, the interpretation of the experimental results and the extent of the claims made.

Reviewer's Responses to Questions

**Part I - Summary**

Reviewer #1: In the present study, Joseph et al. investigate the in vivo role of the HTLV-1-CTCF binding site (vCTCF-BS), using 2 humanized mouse models.

Infection was initiated with 729B cells transfected with: i) HTLV-1-WT, ii) HTLV-1-CTCF, which contains a provirus with a mutated vCTCF-BS, and a stop codon, which deletes the last 23 amino acids of p12, and iii) HTLV-1-p12stop, which contains the intact vCTCF-BS, but retains the same stop codon in p12.

Based on their observations, the Authors suggest a key role of the CTCF binding site in HTLV-1 replication and pathogenicity in vivo.

Reviewer #2: In Joseph et al, the authors assess the role of the HTLV-1 CTCF binding site in viral replication and pathogenesis in vivo using a humanized mouse model. Mutation of the vCTCF-BS delayed virus spread and delayed or abrogated lymphoproliferative disease in HTLV-1 infected Hu-mice. The data further suggest that loss of the vCTCF-BS results in significant effects on host gene expression and expansion of human T cell populations in vivo, including suppression of activity of CD8+ T-cells, and that the vCTCF-BS has dynamic regulation of HTLV-1 gene expression.

Interestingly, previous in vitro studies carried out by the same group revealed that mutation of the vCTCF-BS did not disrupt the kinetics and levels of virus gene expression. Moreover, no effects on minus strand gene expression were seen in vitro.

This is an interesting study for the field. It exploits humanized mouse models to unravel the role of a unique CTCF motif present in the HTLV-1 provirus on the regulation of both the host genome in which it integrates and the viral genome in vivo. A weakness of the study is that the impact of the viral CTCF on Tax expression was assessed in an in vitro model, while there was no evidence for this effect in Hu-mice in vivo.

Reviewer #3: Joseph et al. present an investigation into the role of the viral CTCF binding site in a mutated virus that encodes a truncated p12 protein, in a humanised mouse model of infection and pathogenesis. Authors observe a number of markers that are consistent with reduced pathogenicity upon mutation of the CTCF binding site. The manuscript is very clearly written, and easy to follow. However, there are several confounding factors that make it difficult to interpret the data presented.

**Part II – Major Issues: Key Experiments Required for Acceptance**

Reviewer #1: In general, the manuscript is written in a rather convoluted and, at points, confusing manner.

Although addressing the relevant question of the importance of the CTCF in HTLV-1 infection in vivo, the study is weakened by several drawbacks.

• The p12 stop codon and mutations introduced in the CTCF binding site also change amino acids in the viral regulatory protein p30 (80V to D; 83R to L) and truncate p8. The introduced mutations may also change HBZ non-coding RNA, which is known to be critical for T-cell immortalization. All these consideration make it difficult to pinpoint any of the observed differences solely to the function of the CTCF binding site. Consistent with this concern, in many of the analyses the p12stop virus seems to differ markedly from WT (e.g. Fig. 2).

• Many of the differences are observed between HTLV-1-CTCF and p12 stop, but not WT (see Figs. 1 and 2).

• In Fig. 2C the Bone Marrow plot should show a red bar for p12stop and a green bar for HTLV-1-CTCF. Was bone marrow analyzed in the WT virus group?

• The 2 humanized models yield somewhat different results.

• In the CD133 Hu model the experimental groups’ numerosity is not well matched (WT N=7, CTCF N=6, p12stop N=3). In the CD34 Hu model, the experimental groups’ numerosity is not indicated.

• The scRNAseq image is of extremely poor quality, making it hard to interpret the results. In which cell subset did the Authors see HTLV-1 expression? In which cell types does HTLV-1 persist?

• The Authors suggest that lack of CTCF results in delayed onset/smoldering disease. This statement needs to be validated by testing the expression levels (or function) of CTCF in samples form HC vs. smoldering or acute ATL.

Reviewer #2: (No Response)

Reviewer #3: Major Concerns

● Authors state that the p12-stop mutant is comparable to WT virus in cellular assays. However, throughout this manuscript, there are many instances where the p12-stop mutant behaves differently to the WT virus, seemingly increasing the pathogenicity of HTLV-1. In line 145, authors attribute differences between wt and p12-stop to sampling errors, however a consistent trend seen throughout the manuscript suggests truncating p12 might indeed alter pathogenicity in HuMice, which is confounding. Other studies (PMID: 26929370) mutate the CTCF binding site without truncating p12. This mutant should also be included in the study, to control for both substitutions (as was a concern in previous work), and the truncation.

● Line 172: While it is interesting that vCTCF BS mutant HuMice can be stratified into two groups that correlates with PVL, it is important to present this data much more comprehensively. In Fig. 4F, assignment of each mouse into group 1 or 2 needs to be explicitly marked, and in all panels in Fig 4. Data from WT and P12 controls needs to be included in all the main figure panels, especially as authors state that group 1 mice behave similarly to controls (line 176).

● It would be interesting to know whether mice that stratify into two groups behave differently at early time points. Is the PVL consistently higher throughout the study? There are some datapoints in the vCTCF BS mutant mice that seem more comparable to controls, are these the group 1 mice? Given the variability seen in this condition, readers would benefit from a supplementary table, summarising each mouse used throughout the study, including information such as levels of human CD45+ cells, PVL at each time point, disease status, etc.

● Proceeding to perform scRNA-seq on only group 2 vCTCF BS mutants biases for those with a low PVL and no markers of disease progression, and therefore analysis is confounded, and changes in gene expression or cell populations cannot be attributed to perturbation of CTCF binding alone. RNA-seq should be performed on group 1 samples as well, and should be matched with controls for equivalent PVL from both WT and p12-stop groups, to ascertain the effect of CTCF binding.

● Representative data in Fig S4. is really informative. It looks as though there may be some heterogeneity in HTLV-1 transcript expression, as has been observed before. Is this the same for HTLV-1 expressing cells? The tSNE presented in Fig 5B should be presented for each sample as a supplementary figure and expanded to show the different transcripts (single spliced, double spliced, spliced HBZ) as was done in S10 for ALOX5AP expression, and Fig S4 A and B should be expanded to show genome browser tracks for all 7 samples.

● Although the in vitro data presented in Fig 6. is really compelling, the model in Fig 7. is difficult to address while concerns around RNA-seq, and thus functional effects of abrogating viral CTCF binding in vivo, have not been resolved

**Part III – Minor Issues: Editorial and Data Presentation Modifications**

Reviewer #1: n/a

Reviewer #2: Main comments

Introduction

Tax is expressed intermittently in a small proportion of ATLL cells at any given time [4, 5].

Is this consistently observed in any ATLL case? A proportion of ATLLs carry a tax-deficient provirus, either a mutation in the tax ORF that abolishes the transactivation or oncogenic potential of HTLV-1, or a 5’LTR-deficient provirus that has lost the capacity to express Tax. Did the authors examine nucleotide sequences of such proviruses, and is the CTCF consensus sequence conserved in these proviruses (> 30% of ATLLs).

The authors use two distinct methods for humanizing NBSGW mice, intrahepatic injection of human cord blood CD34+ cells, and intratibial injection of CD133 progenitor cells. They describe the CD34+ model at the end of the introduction and refer to Fig 1A. This figure illustrates both the CD34- and CD133-driven models, CD133 being shown on top. It is confusing for the reader to have this figure in the introduction, while CD133 has not yet been described. We suggest to move the figure from the introduction section to Results, with CD34 on top, and CD133 at the bottom (invert).

The reason for using the CD133 model is not clear. The results (line 122-134), although suggesting decreased pathogenicity of the CTCF infected mice, are obviously different, however the numbers insufficient for statistical analysis.

Fig S3: p12stop infected mice (claimed to be the better control) do also have lower proviral loads. The p-values are estimated for CTCF versus wt-infected animals (3B) not CTCF versus p12stop.

Monitoring disease: how is disease measured during the course of infection before death, are signs of disease only assessed by body weight loss (dropped by 20% or more of their initial body mass), or is this also based on cell counts in the blood? Do the mice die from the disease in some cases (line 166: succumbed from infection) or are they systematically euthanized according to the weight loss criteria indicated above? How do you distinguish between a benign lymphoproliferation and leukemia/lymphoma (malignant disease)? Is it correct that mice do never develop “leukemia” (acute ATLL in human) but only “lymphoma” (lymphoma-subtype ATLL in human)?

Line 99: Mice were then inoculated intraperitoneally with …. An equal number of mice were also infected with HTLV-1 with the same mutation found in the HTLV-1-p12stop virus, as well as an additional mutation that abrogates the vCTCF-BS (HTLV-1-CTCF). Eight litters of mice were used to obtain a sufficient number of mice for statistically valid results, with similar numbers, sexes, and levels of CD34+ cells in mice within each litter allocated for infection with HTLV-1-WT, HTLV-1-p12stop, or HTLV-1-CTCF.

Provide specific numbers for the CD34+ experiment. Numbers are given for the CD133 experiment (Infection with HTLV-1-CTCF (n=6) resulted in delayed onset of lymphoproliferative disease compared to mice infected with HTLV-1-WT (n=7) or HTLV-1-p12stop (n=3) (Figure S1A). For the main model, CD34+, how many animals in each group? Among each group, how many were assessed by scRNA-seq? How many are shown in the figures and how were they selected. Show data of remaining animals in Supplemental material.

Example: How did the authors select the animals (7) examined by scRNA-seq? How many did develop lymphoproliferative disease in each group? In Fig S3, sequencing reads of 4 mice are illustrated in IGV while 7 mice have been sequenced. Do the three other samples have similar transcriptional patterns (provide the number of reads observed for each spicing type if not showing the reads in the figure)?

Role in HTLV-1 pathogenesis:

The stratification in CTCF-1: n=8 and CTCF-2: n=7

Fig 4F illustrates the inverse correlation between PVL at 5 weeks and survival. 14/15 mice are shown, how did the 15th mouse behave? For each of the CTCF mice shown in Fig 4F, indicate to which category it belongs, CTCF-1 or CTCF-2.

scRNA-seq transcriptional profiles: Fig 5B indicates color labels for CD4+, CD8+, CD25+ and TCR+ cells (clustering), it also indicates “T-cells” (blue label in the legend). Clarify how colors are used, and do they overlap? i.e. CD4+ cells are also TCR+ etc … Regarding viral reads in scRNA-seq data, the data were described earlier in the manuscript (line 155 and further). It feels confsing for the reader to have this described at two distinct places in the paper, especially that the first description does not mention the CTCF-2 distinction and that only these mice were used for scRNA-seq analysis.

Transcriptional profiles: T cell clonality was high for all samples as revealed by Gini coefficient and Shanon Diversity index. This quantifies the clonal distribution in terms of TCR and does not discriminate between infected and uninfected T-cell clones. Providing a measure of HTLV-1 integration site distribution by assessing viral clonality using DNA of the same tissues will provide an estimate of the nature and relative integration site abundances even in low PVL animals.

RNA-seq data and Supplementary table: while the TCR sequences are informative, HTLV-1 reads per clone are not a good proxy for the presence of the virus since many cells carrying the virus do not express viral genes, or do express these genes at a low level not detectable via scRNA-seq. Examining HTLV-1 clonality in these cells is technically possible and will define the viral IS landscape. Data can then be combined with TCR clonality (RNA, table) and the number of infected cells extrapolated via PVL values to provide a global T-cell/HTLV-1 clonality picture in CTCF versus p12stop animals.

Role of CTCF on Tax expression

Line 365: … support a potentially new discovery that CTCF regulates TAX in vivo.

There is no evidence for this statement from this study in Hu-mice. scRNA-seq data did not reveal significant differences in Tax expression between CTCF and p12stop infected animals. This assumption is based on in vitro work, which, although elegant and convincing by itself in the Jet model system, does not confirm this effect in vivo. The authors may think about a design to better exploit scRNA-seq, i.e. perform scRNA-seq of the splenocytes (human lymphocytes) collected from the same Hu-mice ex vivo and after in vitro culture, or propose an in vivo experiment to assess this assumption.

In the abstract: Overall, these findings indicate that the vCTCF-BS regulates Tax expression, proviral load, and HTLV pathogenicity in vivo. The effect on Tax expression in vivo was not demonstrated, therefore, this statement should be rephrased.

scRNAseq:

Information in the scRNA-seq paragraph of the Methods section is scarce and should be further explained. What was the viability of the cells used for scRNA-seq and how could the authors differentiate between viable mouse cells and viable human lymphocytes? Custom analysis: what does this mean, the type of analysis, how and why it was performed should be clarified.

Figure 5: B: TSNE plots of a representative data set: clarify. A single animal? How was this animal selected? Line 201: Characteristics of predominant T cell clones, including CD4+, CD8+, Treg, and double CD4+CD8+ clones, varied in each sample. The six remaining datasets should be reported in Supplemental material.

Minor comment:

Line 218: “mice infected HTLV-1-CTCF-2 and HTLV-1-p12stop” -> mice infected WITH HTLV-1-CTCF-2 and HTLV-1-p12stop

Reviewer #3: Minor Issues

● If PVL was undetectable in many mice, particularly those harbouring CTCF mutant virus, how was infection confirmed?

● Authors should state the rationale for using the two complementary methods for generating mice

● Line 60: typo - pr instead of pro

● Lines 75-77: CTCF’s role is oversimplified to the point where it is inaccurate. It primarily functions as an insulator protein, which acts by restraining cohesion mediated loop extrusion, and is associated with both transcriptional activation and repression of cellular genes.

● Line 156: “Interestingly, unlike in infected cells in tissue culture [14], hbz transcripts were the most abundant viral mRNAs in infected splenocytes in vivo” and this is also unlike in individuals with HTLV-1 (PMID: 29062917, PMC4847349) however, it’s worth highlighting that individuals with ATL have higher levels of hbz transcripts, which is consistent with lymphoproliferative disease development seen in the Hu-mice

● More detail is needed for Fig S9A. and S11B legend. Volcano plots are showing log2FC between what conditions?

● Legend for Fig S4: causes confusion to say “BAM files of unmapped reads from a subset of 4 samples on IGV”, convention would be to say reads mapping to the HTLV-1 proviral genome

PLOS authors have the option to publish the peer review history of their article (what does this mean? ). If published, this will include your full peer review and any attached files.

**Do you want your identity to be public for this peer review?** For information about this choice, including consent withdrawal, please see our Privacy Policy .

Reviewer #1: No

Reviewer #2: No

Reviewer #3: No
---

## [Decision Letter · Decision Letter 1]

Dear Lee,

Thank you for submitting the revised manuscript, with the detailed point-by-point response to reviewers. We are pleased to inform you that your manuscript 'Role of the CTCF Binding Site in Human T-Cell Leukemia Virus-1 Pathogenesis' has been provisionally accepted for publication in PLOS Pathogens.

Best wishes,

Charles

Charles R M Bangham, ScD FRS

Academic Editor

PLOS Pathogens

Richard Koup

Section Editor

PLOS Pathogens

Sumita Bhaduri-McIntosh

Editor-in-Chief

PLOS Pathogens

orcid.org/0000-0003-2946-9497

Michael Malim

Editor-in-Chief

PLOS Pathogens

orcid.org/0000-0002-7699-2064

Reviewer Comments (if any, and for reference):

Reviewer's Responses to Questions

**Part I - Summary**

Reviewer #1: The authors have made an appreciable effort to improve the qualityt of their manuscript that is now acceptable for publication.

Reviewer #2: The authors investigated the effects of the vCTCF-BS mutation in vivo using a humanized mouse model. Their findings highlight the critical role of CTCF binding to the HTLV-1 provirus in the dynamic regulation of viral replication and pathogenicity. Although one limitation of the study is that the temporal regulation of Tax expression by CTCF during the early stages of infection was assessed only in tissue culture models and has yet to be confirmed in vivo, this work overall supports a potentially novel discovery that CTCF regulates Tax expression.

The revised manuscript is of high quality. The authors have addressed all my comments thoroughly and have clarified all previously unclear sections. I am satisfied with the new experiments performed, as well as the changes made to the main text and the new supplementary figures.

Reviewer #4: Joseph et al. have provided a revised manuscript of their study into the in vivo role of the viral CTCF binding site, exploring viral replication, pathogenicity and cellular transcription upon mutating vCTCF-BS, alongside measuring the effect of vCTCF-BS on Tax expression in vitro in HEK293T cells. There is a lack of information and inconsistencies that make interpretation of the data, and therefore assessing the quality of this manuscript unfeasible.

**Part II – Major Issues: Key Experiments Required for Acceptance**

Reviewer #1: none

Reviewer #2: -

Reviewer #4: - “Mice from each litter with at least 5% human CD45+ cells, were separated into groups based on levels of human CD45+ cells.”

5% reconstitution is very low, most hu-mice studies (even in non-irradiated mice) are >> 5%. How was this cutoff decided and justified? It has impeded abilities to perform downstream analyses “HTLV-1-CTCF-B mouse could not be analyzed due to lack of adequate numbers of human cells” Reconstitution rates need to be included in the new supplementary tables, as this is important for interpreting results – eg. Are the mice that reconstituted poorly those that form group B?

- Authors state in their revised manuscript:

“Single cell RNAseq analysis was performed on HTLV-1-CTCF-A mice with the highest proviral loads (Figure 5). Sc RNAseq analysis performed on a HTLV-1-CTCF-B mouse could not be analyzed due to lack of adequate numbers of human cells. Levels of human cells in splenocytes of HTLV-1-CTCF-B mice were 4.5-fold lower than those in HTLV-1-CTCF1 mice (Fig 3E).”

However in the first submission, analysis was performed on “3 HTLV-1-CTCF-2 (renamed to group B in the revised manuscript)” mice. Authors need to clarify why why Group 2 (now B) was analysed previously?

In Figure S4 of the first submission CTCF-7 and CTCF-8 were included in group 2 (now B). In the revised manuscript CTCF-8 has been allocated to group A. Is this just a name change? Or have samples been changed? The counts have changed since the first submission, have these data been reanalysed?

In authors response to reviewers "Response: Revised Fig S8 shows the transcripts in each of the 7 animals examined by scRNAseq. Fig S15 shows tSNE plots for each of the 7 animals."

The reviewer requested variation in viral transcripts be presented for each individual animal, not ALOX5AP. This was requested to assess variation in detection of viral transcript expression. This should be included as supplemental information.

"These data suggest that loss of the vCTCF-BS results in significant effects on gene expression and expansion of human T cell populations in vivo."

This concluding statement is vague. Differences in gene expression discussed seem to be selected for discussion as genes of interest, rather than through a robust statistical analysis. Furthermore, the plots in figure S14 are variable and don't support the statements in text. And figure s16 shows only a single representative sample from each condition, again, this is not statistically robust. Either aggregate data for all replicated should be presented, or individual volcano plots for each mouse.

**Part III – Minor Issues: Editorial and Data Presentation Modifications**

Reviewer #1: none

Reviewer #2: -

Reviewer #4: The analyses of the genomics data is unfocused, and unintegrated. There are interesting observations regarding say eg. the mutations acquired in the viral transcripts, and proviral integration site. However these data are mixed within the transcriptomics data, and are not integrated. It would be better to include these in a seperate analyses, not in the middle of the analysis of scRNAseq

(Figure 11B) should read (Figure S11B)

PLOS authors have the option to publish the peer review history of their article (what does this mean? ). If published, this will include your full peer review and any attached files.

**Do you want your identity to be public for this peer review?** For information about this choice, including consent withdrawal, please see our Privacy Policy .

Reviewer #1: No

Reviewer #2: No

Reviewer #4: No

---

## [Editor Report · Acceptance letter]

Dear Dr. Ratner,

We are delighted to inform you that your manuscript, "Role of the CTCF Binding Site in Human T-Cell Leukemia Virus-1 Pathogenesis," has been formally accepted for publication in PLOS Pathogens.

Best regards,

Sumita Bhaduri-McIntosh

Editor-in-Chief

PLOS Pathogens

orcid.org/0000-0003-2946-9497

Michael Malim

Editor-in-Chief

PLOS Pathogens

orcid.org/0000-0002-7699-2064